# When Replanning Becomes the Bottleneck: Budgeted Replanning for Embodied Agents

**Shuaijun Liu** [1]  **Feiyang You** [1]  **Xingwei Chen** [1]  **Ningxin Su** [1 †]

**Project Website:** https://nebulis-lab.com/BRACE

## Abstract

Embodied agents replan frequently to recover from execution drift, partial observability, and coordination hazards, but each LLM-based replanning call can consume an accumulated textual context that grows over time and across agents. Once this context becomes large, replanning latency develops heavy tails and can miss real-time deadlines even when task success remains high, a failure mode that is hard to detect from average latency or success alone. We present BRACE, a controller that formulates replanning as a budgeted control loop by deciding whether to replan, selecting a replanning mode, and allocating an explicit token budget and latency service-level objective (SLO) while accounting for optional efficiency modules. As a reusable component, we introduce E-RECAP, a cost-aware progressive token pruning method that predicts token utility and prunes replanning contexts across transformer layers while preserving critical head and tail tokens. Across Meta Habitat, RoboFactory, and AirSim, BRACE with E-RECAP reduces replanning-call token counts by 62–92% and SLO violation rates from 85.5–100.0% to 4.7–50.0% in settings where task success is already saturated. In a harder RoboFactory setting where open-loop, frozen-plan, and No BRACE all fail, BRACE + E-RECAP reaches 80.0% success with 4.6% SLO violations, demonstrating that tail-aware per-call budgeting is effective across embodied platforms.

†Corresponding author. [1]IoT, Information Hub, The Hong Kong University of Science and Technology (Guangzhou), Guangzhou, Guangdong, China. Correspondence to: Ningxin Su <ningxinsu@hkust-gz.edu.cn>.

*Proceedings of the $43^{rd}$ International Conference on Machine Learning*, Seoul, South Korea. PMLR 306, 2026. Copyright 2026 by the author(s).

## 1. Introduction

Embodied agents operate under partial observability, actuation noise, and changing environments, so long-horizon plans can become invalid after only a few steps (Nasiriany et al., 2024; Feng et al., 2025). Modern systems therefore run an *observe → plan → act → replan* loop and rely on frequent replanning for recovery (Kwon et al., 2025; Wu et al., 2025; Shcherba et al., 2025; Feng et al., 2025). While recent vision-language-action (VLA) and LLM-based planners improve plan quality, the *systems cost* of replanning is rarely treated as a first-class object.

A replanning call is not a lightweight query: prompts include task specification, recent history, failure traces, and, in multi-agent settings, messages and coordination summaries (Chang et al., 2025; Zhang et al., 2025a; Zu et al., 2025; Kang et al., 2025; Li et al., 2025c). As this context grows, transformer planners become slow and bursty; in a closed loop, slow calls delay execution and can trigger more replanning.

Most embodied evaluations emphasize task success or mean latency, which can mask deadline misses. In our experiments, a baseline without budgeting or compression ("No BRACE") achieves 100% success on three platforms while violating the replanning latency service-level objective (SLO) on most replanning calls, showed in Table 2. This motivates treating replanning as a budgeted systems primitive and reporting tail latency and SLO violations at call granularity.

We propose **BRACE** (Budgeted Replanning and Coordination for Embodied Agents), a budgeted replanning systems framework with an explicit controller/accounting interface for each replanning call. At each trigger, BRACE decides whether replanning should occur, selects a replanning mode, and sets a token budget $B_t$ (planner input after compression) together with a latency target $\text{SLO}_t$. To reduce instability from rapid oscillations, BRACE supports cooldown/commit windows and failure-aware overrides that temporarily relax budgets after repeated failures. BRACE is modular and can include context-compression and reuse mechanisms on the replanning call path (e.g., token pruning, retrieval, plan

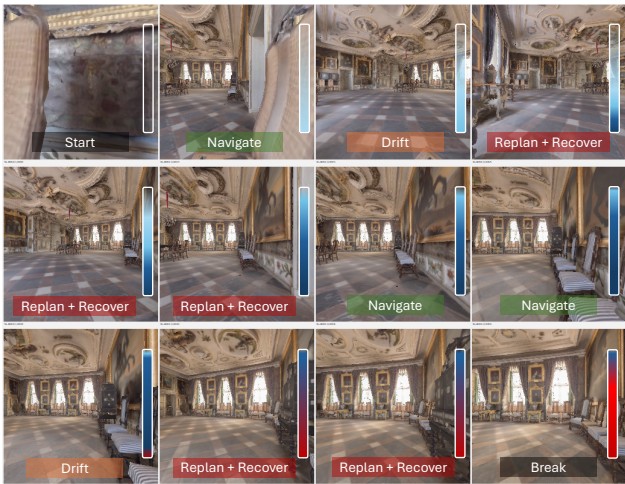

*Figure 1.* Qualitative motivating example on Meta Habitat. As replanning history accumulates, the context grows and produces tail-latency spikes, motivating budgeted context compression.

caching, and communication compression).

To make composition comparable, BRACE instruments the replanning call path with phase accounting and audit logs. For each replanning call, we record tokens and latency for context compression, retrieval, planning, and optional modules, and summarize results with tail percentiles and SLO violation rates. This also enables *budget-matched* baselines, where alternative context-reduction strategies are constrained to the same token budget to separate token count from matched-budget quality retention.

As an efficiency primitive for replanning, we introduce E-RECAP (Embodied REplanning with Cost-Aware Pruning), a progressive token-pruning module that compresses replanning context while preserving critical tokens. E-RECAP is trained to predict token importance and prunes across selected transformer layers, keeping head/tail tokens and selecting the remainder by predicted utility. On Habitat-Lab navigation with multi-agent context growth, E-RECAP reduces tokens per replanning call by 71–76% and replanning latency by 2.1–2.6× with minimal changes in success and success weighted by path length.

**Motivating example: tail latency and SLO violations.** In Meta Habitat navigation with shortest-path-noise execution and a replanning SLO of 2500 ms, No BRACE violates the SLO on 85.5% of replanning calls despite 100% success (Table 2). With E-RECAP on the replanning call path (as used by BRACE), the violation rate drops to 4.7% without reducing success (Table 2 and Appendix Table 15). Figure 1 shows a representative snapshot.

**Contributions.** (1) We show that replanning cost and variability under context growth can dominate closed-loop behavior and that success can hide persistent deadline misses, motivating per-call tail/SLO reporting; (2) introduce BRACE, a budgeted replanning controller with stability policies and phase-level accounting that enables auditable, budget-matched comparisons; and (3) introduce E-RECAP, a trained progressive token-pruning module for replanning contexts, and evaluate it across navigation, manipulation/coordination, and multi-agent traffic benchmarks, including no-replanning baselines, harder-setting evidence, and additional real-robot and cross-platform coverage.

## 2. Related Work

**LLM/VLA planning and recovery.** Recent embodied work strengthens perception-to-action policies and LLM-based reasoning, including VLA models and planner-controller stacks for decomposition, correction, and safety-aware plan generation (Zitkovich et al., 2023; Kim et al., 2025b; Black et al., 2025; Goyal et al., 2025; Zhou et al., 2025; Kwon et al., 2025; Wu et al., 2025; Shcherba et al., 2025; Feng et al., 2025). These methods primarily target plan quality and task completion. Our focus is the systems behavior of *replanning* under long-horizon context accumulation: each planner invocation is treated as a metered replanning call with explicit token and latency budgets. This distinction matters because a planner that is accurate in isolation can still destabilize a closed-loop agent if repeated calls become slow, bursty, or difficult to audit.

**Embodied benchmarks and multi-agent settings.** Modern benchmarks expand long-horizon evaluation across navigation, manipulation, and collaboration (Savva et al., 2019; Szot et al., 2021; Puig et al., 2024; Deitke et al., 2022; Chevalier-Boisvert et al., 2019; Shridhar et al., 2020; Padmakumar et al., 2022; Li et al., 2023; Wang et al., 2025b; Zha et al., 2025; Chang et al., 2025; Zhang et al., 2025a; Zu et al., 2025; Kang et al., 2025; Nasiriany et al., 2024). These settings naturally induce context growth through histories, summaries, and coordination messages. Reported metrics typically emphasize success and sometimes path efficiency, while recent work argues for broader measurement beyond aggregate success (Sohn et al., 2025). We therefore report call-level tail replanning latency, SLO violation rates, and stability proxies such as churn and coordination wait.

**Metareasoning and resource-bounded planning.** Classical metareasoning studies how an agent should allocate computation by weighing the value of additional reasoning against its cost, providing a foundation for resource-bounded rationality (Russell & Wefald, 1991; Lin et al., 2015). This connection is direct in our setting because replanning competes with acting for wall-clock time. Our contribution is not an optimal metareasoning solution for a known MDP/POMDP model; instead, we expose an embodied-systems interface with explicit token/SLO bud-

gets, phase accounting, and empirical stability metrics for LLM/VLA replanning pipelines.

**Efficiency, caching, retrieval, and token pruning.** System work on VLA efficiency often targets per-step inference via caching, adaptive reuse, scheduling, or pruning (Xu et al., 2025; Li et al., 2025b). Related token-pruning work typically focuses on visual-token pruning or single-step queries (Zhang et al., 2025b; Liu et al., 2025; Kim et al., 2025c; Li et al., 2025a). Our setting differs in both object and timing: E-RECAP prunes *replanning context* on the trigger-driven replanning call path. Retrieval-based planning and memory reuse are also actively explored (Guo et al., 2025; Ling et al., 2025; Cui et al., 2025); BRACE composes with retrieval and caching while making their overhead auditable via phase accounting, enabling budget-matched comparisons rather than attributing all cost changes to the planner alone.

## 3. Method

### 3.1. Replanning Under Context Growth and Deadlines

We study embodied agents that operate in a closed loop: the agent executes a plan, observes the resulting state, and revises the plan when execution drifts, failures occur, or safety/coordination signals are raised. In LLM/VLA-based systems, a replanning request typically carries a large textual prompt including task specification, recent observations/actions, failure traces, and, in multi-agent settings, messages or summaries from other agents. We denote this accumulated replanning context at controller step $t$ by $C_t$ and its token length by $N_t$. As episodes progress and more agents contribute, $N_t$ increases and replanning becomes slower and more variable.

BRACE treats each *potential* replanning point as a controller decision. At step $t$, the controller decides whether to invoke the planner. If it does, we call that a *replanning call*. A replanning call is not just the planner forward pass: it may include context compression, retrieval, or clarification. We measure and enforce budgets on this end-to-end call path.

**Budgets and the replanning call path.** BRACE exposes two per-call budget variables: (1) a *token budget* $B_t$, defined as the number of tokens passed into the planner *after* any compression or budgeting, and (2) a *latency target* $\mathrm{SLO}_t$ (ms), interpreted as a deadline for the end-to-end replanning call latency (including enabled modules). Our accounting and evaluation are therefore call-level: we report tokens, latency percentiles, and deadline-miss rates across replanning calls rather than only episode-level aggregates. Figure 2 summarizes the BRACE controller and the modules that can appear on the replanning call path.

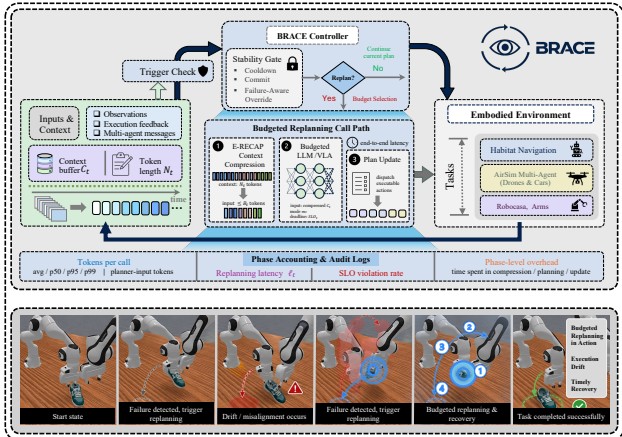

*Figure 2.* Overview of BRACE: a closed-loop controller that determines whether to invoke replanning and selects the per-call token budget and latency SLO, with E-RECAP as a composable token-pruning module on the replanning call path.

### 3.2. BRACE Controller: When to Replan and How Much to Spend

BRACE maps trigger signals and a small controller state to replanning decisions and per-call budgets. It has two responsibilities: (1) decide whether a trigger should result in a planner invocation, and (2) if replanning is executed, choose the replanning mode and budgets.

Let $\tau_t$ denote a trigger indicator (or vector of trigger flags) at step $t$. Triggers can be periodic, failure-driven (e.g., repeated action failures or divergence from an expected trajectory), or hazard-driven (e.g., safety or coordination signals in multi-agent settings). BRACE does not hard-code a single "always replan on trigger" rule. Instead, triggers are inputs to a controller that can skip replanning, replan with a constrained budget, or temporarily relax budgets after repeated failures.

**Controller state and stability policies.** BRACE maintains a controller state $\sigma_t$ that summarizes short-horizon execution and replanning history. In practice, $\sigma_t$ includes counters and flags such as the number of steps since the last replanning call, the number of steps since the last plan change, and counts of consecutive failures. These variables support stabilization policies that reduce rapid back-to-back replanning, which can otherwise amplify latency spikes and coordination delays.

We implement two simple stabilization mechanisms: *cooldown* and *commit* windows. Intuitively, cooldown prevents the controller from calling the planner again immediately after a recent call. Commit prevents the controller from discarding a newly produced plan before it has been executed for a minimum horizon. BRACE also supports *failure-aware overrides*: when failures persist, the controller

can bypass or relax these gates and increase budgets to promote recovery.

**A minimal formalization.** Let $\Delta_t$ and $\kappa_t$ be counters encoded in $\sigma_t$: $\Delta_t$ is the number of steps since the last replanning call (cooldown counter), and $\kappa_t$ is the number of steps since the last plan update (commit counter). Let $\delta$ and $\omega$ be the corresponding thresholds. BRACE first applies a stability gate

$$u_t = \mathbb{I}[\tau_t \wedge (\Delta_t \geq \delta) \wedge (\kappa_t \geq \omega)], \tag{1}$$

where $u_t = 1$ means replanning is allowed at step $t$ (subject to any failure-aware override rules in $\sigma_t$). If $u_t = 0$, the agent continues executing the current plan without invoking the planner. Absent failure-aware overrides, Eq. (1) also implies a deterministic anti-churn property: consecutive replanning calls must be separated by at least $\delta$ controller steps, and a newly changed plan must survive at least $\omega$ steps before the gate allows another plan replacement. Appendix B states this property explicitly and gives a short proof sketch. We use it as controller-side structure and empirical auditability, rather than presenting it as a full theorem-level stability guarantee.

If $u_t = 1$, BRACE selects a replanning mode and budgets:

$$(m_t, B_t, \text{SLO}_t) = f(\tau_t, \sigma_t), \tag{2}$$

where $m_t$ chooses which call-path modules are enabled and which planner prompt template is used. The token budget $B_t$ constrains the planner input length after any compression, and $\text{SLO}_t$ specifies the latency target for the end-to-end replanning call.

Explicit token budgets make comparisons cleaner. Given a target $B_t$, any context-reduction baseline (e.g., recency truncation, random truncation, or structured summaries) can be forced to satisfy the same planner-input constraint. This isolates effects due to budgeting and selection from effects due to simply spending more tokens.

### 3.3. E-RECAP: Pruning for Replanning Context

BRACE is designed to compose with modules that reduce replanning cost or reuse prior computation: (1) token pruning (E-RECAP) to compress long contexts before the planner is called, (2) retrieval to reuse prior subplans or failure cases (RAG), (3) caching of previously generated plans or subplans, and (4) optional compression of coordination messages in multi-agent settings. These modules can reduce planner input size but also add latency, so BRACE treats the entire replanning call path as the object of budgeting and measurement and attributes overhead via phase accounting (Section 3.4).

E-RECAP is a context-compression module that enforces a planner-input token budget by pruning the replanning con-

text. The key difference from common VLM/VLA pruning settings is the object being pruned: E-RECAP targets the *long-horizon replanning prompt* accumulated over time and across agents, rather than per-frame visual tokens or single-step queries.

**Context growth model.** In a multi-agent setting with $K$ agents, a simple decomposition of context length is

$$N_t = N_0 + \sum_{i=1}^{t} \sum_{a=1}^{K} n_{i,a}, \tag{3}$$

where $n_{i,a}$ is the number of tokens contributed by agent $a$ between controller steps $i - 1$ and $i$ (e.g., new messages, summaries, or execution traces). When the planner is a transformer, the compute associated with attention grows superlinearly in $N_t$ (often approximated as $\mathcal{O}(N_t^2)$), motivating direct constraints on the planner input length.

**Progressive pruning across layers.** Let the tokenized replanning context be a sequence $x_{1:N}$. E-RECAP prunes progressively at a set of pruning layers $\mathcal{P}$ inside the transformer. At a pruning layer $l \in \mathcal{P}$ with current sequence length $N_l$, let $h_i^{(l)} \in \mathbb{R}^d$ be the hidden state for token $i$. A lightweight predictor produces an importance score $\pi_i^{(l)}$ from $h_i^{(l)}$ (training details are provided in the appendix; see Appendix Table 8).

Given a per-layer keep ratio $r_l \in (0, 1]$, E-RECAP sets the next-layer length

$$N_{l+1} = \lfloor r_l N_l \rfloor. \tag{4}$$

To form the kept index set, E-RECAP always preserves a small set of head tokens and a tail window (to retain task specification and recent context), and fills the remaining budget with top-scoring tokens:

$$\begin{aligned} \mathcal{I}_{l+1} = \mathcal{H}_l &\cup \mathcal{T}_l \\ &\cup \text{TopK}\Big( \{\pi_i^{(l)}\}_{i=1}^{N_l} \setminus (\mathcal{H}_l \cup \mathcal{T}_l), \\ &N_{l+1} - |\mathcal{H}_l| - |\mathcal{T}_l| \Big), \end{aligned} \tag{5}$$

and the sequence is pruned by restricting to indices in $\mathcal{I}_{l+1}$. After the final pruning layer, the resulting token sequence is used as the planner input and is constrained to satisfy the per-call token budget $B_t$ (up to rounding effects from the discrete selection). Figure 3 shows where E-RECAP sits on the replanning call path; pruning schedules, training objectives, and training-data mixtures (including Dolly/Alpaca/Self-Instruct and embodied auxiliary data such as ALFRED/TEACh/BabyAI) are provided in the appendix (Appendix Table 8 for pruning configuration and Appendix Table 28 for the data-mixture sweep).

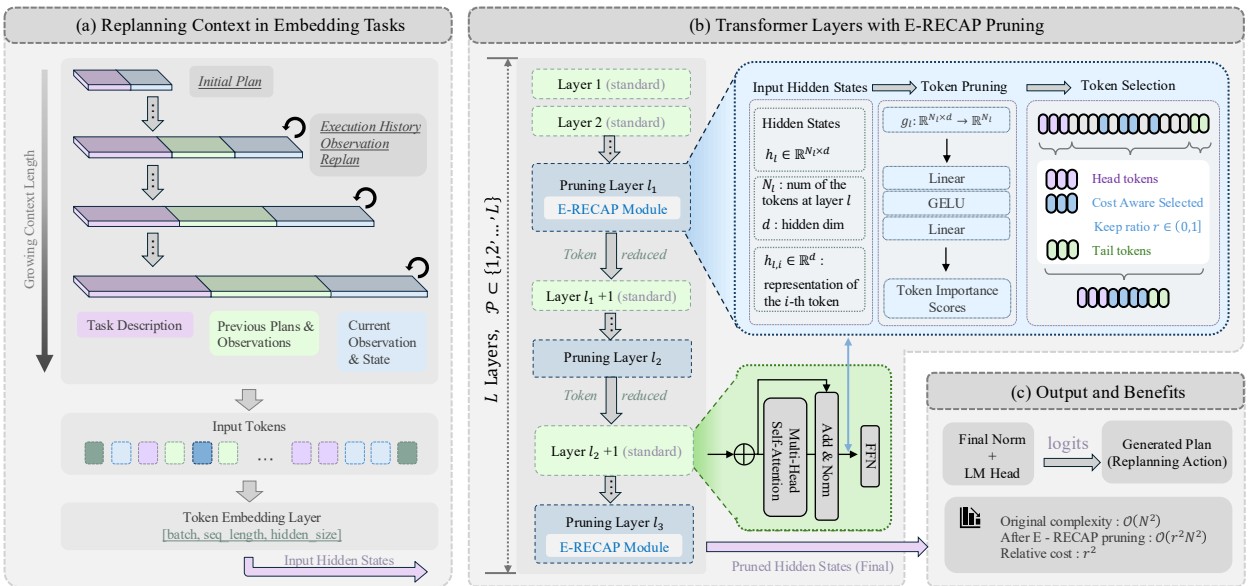

*Figure 3.* E-RECAP module. A lightweight predictor scores context tokens using intermediate hidden states, and pruning is applied progressively at selected layers. The kept set includes fixed head and tail tokens plus top-scoring tokens, producing a shorter context that satisfies the planner-input budget.

### 3.4. Phase Accounting and Audit Logging

BRACE instruments the replanning call path with phase-level accounting. For each controller step $t$, we log timestamps and token counts at phase boundaries. When $u_t = 1$ and a replanning call is executed, we measure the end-to-end replanning latency $\ell_t$ as the wall-clock time from entering the replanning call path to returning a plan (including any enabled modules). We also log per-phase latencies (e.g., pruning time, retrieval time, planner time, clarification time) so that overhead is not implicitly attributed to the planner.

We define the *SLO violation rate* as the fraction of replanning calls that miss the latency deadline:

$$v = \frac{1}{T} \sum_{t=1}^{T} \mathbb{I}[\ell_t > \text{SLO}], \qquad (6)$$

where $T$ is the number of replanning calls and $\mathbb{I}[\cdot]$ is an indicator. Table 1 summarizes the main accounting quantities used in the paper.

**What gets logged.** A replanning call is logged as an ordered sequence of phases (e.g., context compression, retrieval, planner invocation, optional clarification, and execution dispatch). Each phase records its own token counts and timing. This design supports two uses: (1) reporting tail latency and SLO violations for the end-to-end replanning call, and (2) attributing tail events to concrete sources (e.g., retrieval overhead versus planner latency), rather than collapsing all effects into one end-to-end number.

*Table 1.* Notation and metrics used for replanning accounting. The table defines the per-call context, token budget, latency SLO, trigger decision, and diagnostic quantities used throughout the experiments.

| Symbol | Type | Description |
|---|---|---|
| $t$ | integer | Controller step within an episode; replanning occurs when $u_t{=}1$. |
| $K$ | integer | Number of agents. |
| $B$ | integer | Per-call planner-input token budget (after compression). |
| $r$ | scalar | Per-layer keep ratio (E-RECAP); overall reduction is measured. |
| $\ell_t$ | scalar | End-to-end replanning latency for call $t$ (ms). |
| $\text{P95}(\ell)$ | scalar | 95th percentile of replanning latency (tail). |
| SLO | scalar | Latency deadline (ms), domain-specific. |
| $v$ | scalar | SLO violation rate across replanning calls (Eq. (6)). |
| $N_{\text{in}}$ | scalar | Mean tokens before compression (raw context length). |
| $N_{\text{after}}$ | scalar | Mean tokens passed into the planner (after compression). |

## 4. Experiments

Our evaluation emphasizes (i) tail replanning latency and SLO violations under context growth, (ii) budget-matched comparisons that isolate budgeting and selection effects beyond raw token count, and (iii) composability on the replanning call path (e.g., pruning and retrieval) with auditable phase-level overheads.

**Algorithm 1** High-level BRACE controller for budgeted replanning. The pseudocode shows trigger admission, budget selection, optional compression, planner invocation, and audit logging on the replanning call path.

---

**Require:** Environment state $s_t$, context buffer $C_t$, trigger signals $\tau_t$, controller state $\sigma_t$ (cooldown/commit/failure counters), global budget bounds $(B, \text{SLO})$

**Ensure:** Executed action(s) or updated plan $\pi_t$; updated controller state $\sigma_{t+1}$; phase-level audit logs

1: **Sense:** append observation to $C_t$; log `sense`
2: **Trigger eval:** compute $\tau_t$; update $\sigma_t$; log `trigger_eval`
3: **Stability gate:** compute $u_t$ via Eq. (1) (failure-aware overrides); log `stability_gate`
4: **if** $u_t = 0$ **then**
5:     Execute current plan; update $\sigma_{t+1}$; **return**
6: **end if**
7: **Budget select:** choose $(m_t, B_t, \text{SLO}_t)$ via Eq. (2); log `budget_select`
8: **Context compress (optional):** prune/summarize to satisfy $B_t$; log `context_compress`
9: **Retrieve (optional):** retrieve memory (RAG/cache) and merge; log `retrieve`
10: **Replan:** call planner with budgeted context; log `replan`
11: **Execute:** dispatch $\pi_t$; log `execute`
12: **Accounting:** update $\sigma_{t+1}$ using outcomes and measured latency; emit audit logs

---

## 4.1. Setup

**Platforms and scenarios.** Our evaluation spans navigation, manipulation/coordination, and multi-agent traffic/UAV interaction. The main-text simulation anchors are Meta Habitat navigation (Savva et al., 2019; Szot et al., 2021; Puig et al., 2024), RoboFactory manipulation/coordination (Qin et al., 2025), and Microsoft AirSim multi-agent scenarios (Shah et al., 2018). We additionally include manipulation extensions on RoboSuite and LIBERO and summarize the broader simulation coverage in Appendix Table 12. In AirSim, $K$ denotes the number of agents (here $K = 8$ for the intersection benchmark). Across all platforms, we treat a *replanning call* as the unit of accounting. In RoboFactory, low-level execution uses an external executor family (OpenMARL; e.g., OpenVLA / Pi0 / DP); when invoked, we account its overhead as part of the execution phase rather than attributing it to the replanning call.

**Metrics.** We report task success alongside replanning cost measured as (i) tokens passed into the replanning call *after* any compression/budgeting and (ii) end-to-end replanning latency, including enabled modules on the replanning call path (e.g., pruning, retrieval). We summarize tail behavior

with percentiles (P95/P99) and report the SLO violation rate (fraction of replanning calls with $\ell_t > \text{SLO}$). Unless otherwise noted, values are rounded from the evaluation summaries. Experiments run on a server with $8\times$ NVIDIA RTX 5880 Ada GPUs (48GB VRAM each); we use single-GPU mode unless otherwise noted.

Per-domain setup parameters, variant tables, and additional metrics are reported in the appendix (Appendix Table 11 and Appendix Tables 18–24), including the Habitat clarification axis, the controlled controller sweep, and additional AirSim/RoboFactory ablations. We keep these details in the appendix to preserve a compact main-paper narrative while making the evidence traceable under the same accounting definitions.

## 4.2. Baselines

We use **No BRACE** to explicitly denote a baseline with no budgeting and no compression on the entire replanning call path (no pruning, no retrieval, no caching, no communication compression). we also report true no-replanning conditions (e.g., no-initial-plan / open-loop) and fixed frozen-plan conditions as system baselines rather than selector baselines. To better isolate the effect of budgeting and accounting (rather than extra compute), we include baselines that enforce comparable token budgets through alternative context reduction strategies: *random truncation*, *recency truncation*, and *structured summaries*. We additionally include learned, retrieval-augmented, and recent top-venue alternatives when they strictly share the same accounting contract and budget. These baselines match a planner input budget $B$ without using BRACE's controller policies, enabling fair comparisons beyond raw token count. Table 3 reports the selector-side and recent-method comparison as a full-width table, while Table 4 keeps the original heuristic-only budget-matched baseline table for full traceability.

## 4.3. Results

**Cross-platform snapshot.**

Table 2 shows two consistent patterns: success can saturate while replanning repeatedly misses deadlines, and reducing replanning tokens is necessary but not sufficient. BRACE's budgeting and accounting make tail behavior explicit, and composable modules such as E-RECAP reduce both tail latency and SLO violations. AirSim still exhibits a heavy tail, so we emphasize violation rates and tail percentiles rather than averages alone. The additional evidence is cross-domain rather than Habitat-only: Habitat contributes no-replanning, phase-overhead, and direct-effect slices; RoboFactory contributes the harder-setting/open-loop separation; and AirSim contributes trigger-audit and safety-proxy evidence in the appendix (Appendix Tables 25 and 24).

*Table 2.* Cross-platform summary of replanning cost and stability. "Tokens" denotes the mean number of tokens passed into the replanning call after pruning/budgeting; tail latency is the P95 replanning latency; SLO violation is the fraction of replanning calls that exceed the per-domain latency SLO. **No BRACE** denotes the configuration without budgeting and without pruning (no compression). Shaded rows highlight No BRACE and BRACE(+E-RECAP).

| Platform | Scenario | Method | Ep | Success (%) | Tokens | Lat P95 (ms) | SLO (ms) | SLO viol (%) |
|---|---|---|---|---|---|---|---|---|
| Meta Habitat | Navigation | No BRACE | 30 | 100.0 | 235 | 2,677 | 2,500 | 85.5 |
| Meta Habitat | Navigation | **BRACE + E-RECAP** | 30 | 100.0 | **20** | **2,500** | 2,500 | **4.7** |
| RoboFactory | Pass-Shoe manipulation | No BRACE | 10 | 100.0 | 1,566 | 1,604 | 250 | 100.0 |
| RoboFactory | Pass-Shoe manipulation | **BRACE + E-RECAP** | 10 | 100.0 | **319** | **1,213** | 250 | **50.0** |
| Microsoft AirSim | $K{=}8$ intersection | No BRACE | 10 | 100.0 | 2,934 | 8,520 | 2,500 | 100.0 |
| Microsoft AirSim | $K{=}8$ intersection | **BRACE + E-RECAP** | 10 | 100.0 | **1,114** | **1,640** | 2,500 | **4.7** |

**RoboFactory budget-matched methods on the shared anchor.** Table 3 restores the wide selector-side and recent-method comparison on the Pass-Shoe anchor. Table 4 reports the heuristic-only budget-matched baselines under the same accounting contract, preserving the original strongest-heuristic comparison without folding the external-method rows into a different panel layout.

**E-RECAP replanning acceleration under context growth.** Table 5 reports E-RECAP's per-replanning-call acceleration on Habitat-Lab (MP3D) as the number of agents $K$ increases. This directly supports the efficiency claim used throughout the paper: under multi-agent context growth, E-RECAP removes 71–76% of tokens per replanning call and yields 2.1–2.6× replanning-latency speedups with minimal changes in success or SPL. Training-data configuration effects and backbone robustness are summarized in Appendix Tables 28 and 27; we keep the main text focused on the call-level efficiency result.

**Meta Habitat variants and tail distributions.** The full Meta Habitat variant table is reported in Appendix Table 15.

In Meta Habitat with shortest-path-noise execution, pruning dominates the improvement: both BRACE+E-RECAP and No-BRACE+E-RECAP reduce the replanning input from ≈235 tokens to ≈20, cutting SLO violations from ≈85% to single digits. At the same budget $B{=}20$, different budget-matched truncations behave differently: 'Recency' is the strongest heuristic in this matched-budget regime and also reaches 0.0% SLO violation, while random/structured reductions are substantially weaker. This is exactly why selector comparisons should be read under matched-budget quality retention rather than raw token count alone. Appendix Tables 16 and 17 further separate phase overhead from task quality: on the Habitat slice, E-RECAP adds only ≈ 35 ms pruning overhead relative to ≈ 2.4 s planner latency while preserving task quality and restoring schedulability.

Figure 4 summarizes distribution-level tail behavior: (a) a CDF with the SLO threshold (Meta Habitat) and (b) a cross-platform SLO violation snapshot. Why it matters: tail percentiles and violation rates directly correspond to

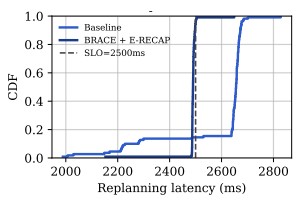

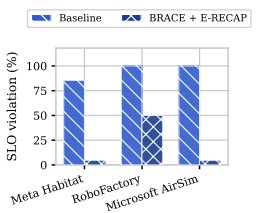

*(a)* Tail latency on Meta Habitat: CDF with the SLO threshold.

*(b)* Cross-platform SLO violation rates.

*Figure 4.* Tail-latency distributions and SLO violation rates across platforms.

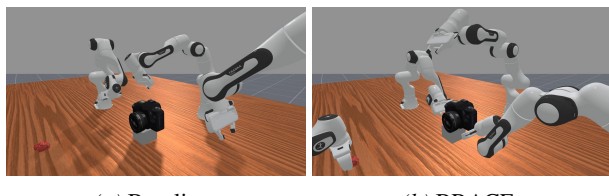

*(a)* Baseline.   *(b)* BRACE.

*Figure 5.* Qualitative example on RoboFactory TakePhoto (multi-agent), showing representative frames from the baseline and BRACE rollouts. Quantitative comparisons are reported in Table 2, and the corresponding ablations in Table 20.

real-time stability, where rare latency spikes can destabilize closed-loop control even when average success is high.

**Qualitative examples.** The representative qualitative comparisons to ground the system-level metrics in observable behavior. Figure 5 keeps the original RoboFactory qualitative example, while Figure 6 uses an AirSimNH storyboard to make trigger-window differences easier to inspect than a single side-by-side frame. Appendix Figure 13 provides a RoboSuite TWO-ARM PEG-IN-HOLE failure-case extension beyond the original RoboFactory anchor. These snapshots help connect tail/SLO metrics to observable coordination outcomes, complementing the aggregated evidence in Table 2.

### 4.4. Ablations

**Open-loop baseline and harder-setting evidence.** Table 6

*Table 3.* Budget-matched comparison of selector-side and recent methods on the RoboFactory Pass-Shoe anchor ($B = 128$). All methods listed here achieve task success; we therefore omit the Success column and report cost metrics only.

| Method | Lat P95 | SLO viol. | Tok after | Wait | Type |
|---|---|---|---|---|---|
| **E-RECAP** | 207.09 | 0.49% | **125.66** | **5776.50** | learned pruning |
| **KVTC** (Staniszewski & Łańcucki, 2026) | 210.33 | 1.01% | 125.86 | 5979.35 | quantization |
| **Robust Compression Boundary** (Yu et al., 2025) | 210.33 | 4.0% | 125.86 | 5979.35 | compression |
| **TurboQuant** (Zandieh et al., 2026) | 221.57 | 1.73% | 125.90 | 6021.51 | quantization |
| **Cross Distillation Compression** (Wang et al., 2025a) | 221.57 | 7.0% | 125.90 | 6021.51 | distillation |
| **ReST-KV** (An et al., 2026) | 231.95 | 2.27% | 125.81 | 6807.52 | selector |
| **Sub-LIME** (Saranathan et al., 2025) | 231.95 | 9.0% | 125.81 | 6807.52 | selector |
| **DefensiveKV** (Feng et al., 2026) | 239.09 | 3.49% | 125.83 | 6862.33 | selector |
| **Gated Attention** (Qiu et al., 2025) | 239.10 | 14.0% | 125.83 | 6862.33 | selector |
| Grad-Hidden Saliency | 239.67 | 2.43% | 125.98 | 7079.05 | derived selector |
| **FreeKV** (Liu et al., 2026) | 239.99 | 3.47% | 126.64 | 7269.27 | retrieval |
| **Token Recycling** (Luo et al., 2025) | 239.99 | 14.0% | 126.64 | 7269.27 | recycling |

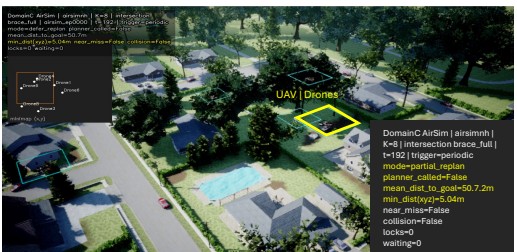
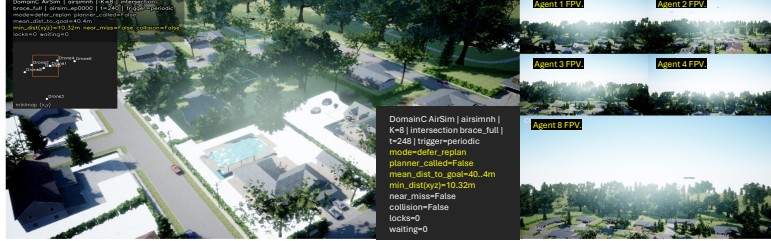

*Figure 6.* Qualitative comparison on AirSimNH. The baseline accumulates delayed replanning decisions within the same trigger window, whereas BRACE shortens the effective replanning path and completes the interaction with fewer deadline misses. The corresponding quantitative results are reported in Table 2 and Appendix Table 10.

*Table 4.* Budget-matched heuristic baselines on the RoboFactory Pass-Shoe anchor ($B = 128$), reported under the same accounting contract as Table 3.

| Method | Tokens | Lat P95 | Lat P99 | Bind rate (%) |
|---|---|---|---|---|
| Random truncation | 128 | **200.8** | 239.6 | 94.9 |
| Recency truncation | 128 | 204.9 | **225.6** | **95.0** |
| Structured summary | 128 | 314.4 | 347.3 | 94.2 |

*Table 5.* E-RECAP replanning acceleration on Habitat-Lab (MP3D) PointNav at $r = 0.7$. The ObjectNav extension is reported in Appendix Table 13.

| Task | K | Method | Keep Ratio | Success | SPL | Tokens/Replan | Latency (s) | Speedup | Token Reduction |
|---|---|---|---|---|---|---|---|---|---|
| | | | | *PointNav Task* | | | | | |
| | 1 | No-Pruning | 1.0 | 0.85 | 0.72 | 2,847 | 2.34 | 1.00× | 0% |
| | | Random | 0.7 | 0.78 | 0.65 | 823 | 1.12 | 2.09× | 71.1% |
| | | **E-RECAP** | **0.7** | **0.84** | **0.71** | **823** | **1.12** | **2.09×** | **71.1%** |
| PointNav | 4 | No-Pruning | 1.0 | 0.84 | 0.71 | 12,847 | 9.87 | 1.00× | 0% |
| | | Random | 0.7 | 0.65 | 0.57 | 3,421 | 4.23 | 2.33× | 73.4% |
| | | **E-RECAP** | **0.7** | **0.83** | **0.70** | **3,421** | **4.23** | **2.33×** | **73.4%** |
| | 8 | No-Pruning | 1.0 | 0.80 | 0.67 | 38,924 | 29.67 | 1.00× | 0% |
| | | Random | 0.7 | 0.65 | 0.58 | 9,234 | 11.23 | 2.64× | 76.3% |
| | | **E-RECAP** | **0.7** | **0.79** | **0.66** | **9,234** | **11.23** | **2.64×** | **76.3%** |

*Table 6.* Summary of open-loop and harder-setting comparisons. The Habitat rows contrast no-replanning with budgeted repeated replanning; the RoboFactory rows contrast open-loop, frozen-plan, and BRACE-based recovery under the harder Pass-Shoe setting.

| Domain | Method | Task metric | Cost metric | Lat P95 | SLO viol. |
|---|---|---|---|---|---|
| Habitat | No-initial-plan | 53.3% / 0.519 SPL | 0 replans | N/A | N/A |
| Habitat | No BRACE | 53.3% / 0.515 SPL | 4.533 replans/ep | 5492 | 93.4% |
| Habitat | BRACE + E-RECAP | 53.3% / 0.519 SPL | 4.533 replans/ep | 2486 | 0.0% |
| RoboFactory | Open-loop | 0.0% success | 0 wait | N/A | N/A |
| RoboFactory | Frozen plan | 0.0% success | 55.6 ms wait | 72.8 | 0.0% |
| RoboFactory | No BRACE | 0.0% success | 6607.3 ms wait | 312.7 | 27.6% |
| RoboFactory | BRACE + E-RECAP | 80.0% success | 6241.7 ms wait | 247.2 | 4.6% |

separates two settings. On the Habitat PointNav slice, the true no-replanning baseline shows that this easy regime is mainly a schedulability comparison: repeated replanning increases call-path cost far more than task quality. On RoboFactory Pass-Shoe harder-setting, the distinction is much sharper: open-loop, frozen-plan, and No BRACE all fail, while BRACE+E-RECAP preserves both recovery and deadline control. The older RAG×prune ablation remains available in Appendix Table 20.

**Focused real-robot evaluation.** To complement the simulation-based evidence with a deployment-facing check, we include a focused single-arm real-robot evaluation on two manipulation tasks: PICKFRUIT and PUSHT. We treat this package as a compact validation of the same budgeting/replanning interface in physical execution, rather than as a claim of full deployment completeness. Table 7 reports the compact summary, Figure 7 shows a PICKFRUIT rollout pair on a banana instance, and the appendix reports the detailed setup (Appendix Table 31), per-task failure statistics (Appendix Table 32), and mechanism-facing notes (Appendix C).

**Diagnostics and stability.** Figure 8 provides a compact diagnostic view across domains, and Appendix Table 19 reports additional stability metrics for RoboFactory. Why it matters: this diagnostic view makes the cost–stability trade-offs auditable and comparable across domains, rather than attributing instability solely to task difficulty. In RoboFactory, success saturates quickly, so we report coordination

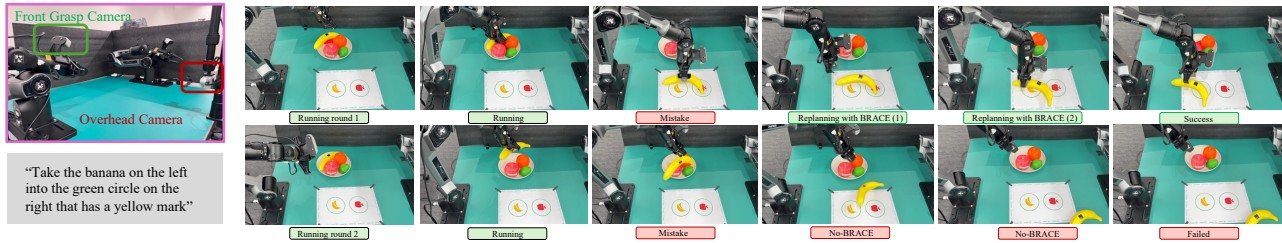

*Figure 7.* Real-robot PICKFRUIT rollout pair on a banana instance. The upper rollout uses BRACE and triggers replanning twice after an intermediate execution failure, subsequently recovering and completing the task; the lower rollout relies solely on the underlying LLM+VLA stack and fails to recover from a comparable execution failure.

*Table 7.* Focused single-arm real-robot results on PICKFRUIT and PUSHT. The summary indicates that the same budgeting and replanning interface remains effective under physical execution.

| Task | Method | Succ. Rate | Replans/Ep | P95 Replan (s) | SLO Viol. |
|---|---|---|---|---|---|
| PICKFRUIT | One-Shot | 8.0% | 0.0 | N/A | N/A |
| PICKFRUIT | No BRACE | 24.0% | 3.4 | 29.4 | 42.7% |
| PICKFRUIT | **BRACE + E-RECAP** | **40.0%** | **1.9** | **22.8** | **18.6%** |
| PUSHT | One-Shot | 0.0% | 0.0 | N/A | N/A |
| PUSHT | No BRACE | 12.0% | 3.8 | 34.7 | 61.3% |
| PUSHT | **BRACE + E-RECAP** | **32.0%** | **2.2** | **26.1** | **27.5%** |

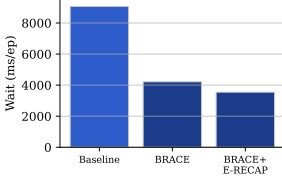

*(a)* Coordination wait (RoboFactory).

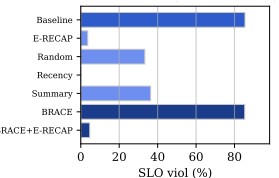

*(b)* SLO violation (Habitat).

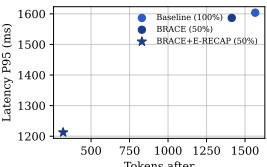

*(c)* Token-latency tradeoff (RoboFactory).

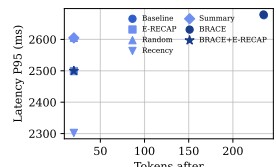

*(d)* Token-latency tradeoff (Habitat).

*Figure 8.* Diagnostic views of coordination wait, SLO violations, and token–latency tradeoffs across the RoboFactory and Habitat domains.

wait time and SLO violations as additional stability signals. Appendix Table 19 shows that budgeting and token pruning reduce both replanning overhead and coordination wait, consistent with the qualitative snapshots in Figure 5. Token budgets are necessary but not sufficient; BRACE's controller and phase accounting make these tradeoffs explicit.

## 5. Discussion

Success can remain high even when replanning routinely misses real-time deadlines, so we emphasize tail latency and SLO violation rates in addition to averages (Sohn et al., 2025), which connects to classical metareasoning and resource-bounded rationality, where computation is treated as a decision variable (Russell & Wefald, 1991; Lin et al., 2015). We frame BRACE as a budgeted replanning systems framework with explicit controller/accounting semantics rather than a theorem-level analysis of the full closed loop. BRACE makes token/latency budgets explicit, supports phase-level cost attribution, and enables budget-matched comparisons when composing modules (e.g., retrieval and pruning), so improvements do not come at the expense of tail behavior. We thus recommend reporting the SLO, at least one tail percentile (e.g., P95/P99), and the violation rate, together with a brief phase-level breakdown when possible. Finally, pruning alone does not prevent feedback loops triggered by repeated failures or coordination hazards; controller policies such as cooldown/commit windows and failure-aware overrides help reduce churn while preserving responsiveness.

## 6. Conclusion

We introduced BRACE, a budgeted replanning framework that treats replanning as a systems problem. Across embodied platforms, explicit budgeting and tail-aware reporting remain necessary because success can stay high while replanning frequently violates latency SLOs under context growth. By composing with efficiency modules such as E-RECAP pruning and retrieval, BRACE reduces replanning tokens and stabilizes tail latency and violation rates. Empirically, across Meta Habitat, RoboFactory, and AirSim this combination cuts SLO violation rates by more than an order of magnitude, and a focused real-robot evaluation on PICKFRUIT and PUSHT corroborates the simulation evidence under physical execution. Several directions remain open, including richer controller policies (e.g., risk-aware budget allocation and failure-conditioned overrides), tighter integration with retrieval and caching backends, and broader coverage of long-horizon manipulation and human-in-the-loop settings. We hope future embodied evaluations report tail-aware replanning cost at call granularity alongside task success (Kim et al., 2025a).

## Impact Statement

This work improves the efficiency and real-time reliability of embodied-agent replanning by treating token usage and latency as explicit, auditable budgets at the granularity of each replanning call. By reporting tail latency and SLO violation rates alongside task success, BRACE encourages embodied evaluations to expose deadline-miss behavior that would otherwise be hidden under aggregate success metrics, which we view as a positive step for the reproducibility and safety reviewability of LLM/VLA-driven robotic systems. Positive impacts include more reliable real-time control loops and more efficient use of computational resources, which may lower the energy and deployment cost of interactive embodied agents, particularly in latency-sensitive settings such as multi-agent coordination, autonomous navigation, and dexterous manipulation. As with many techniques that accelerate autonomous decision-making, the same efficiency gains could in principle be misused to make surveillance, manipulation, or weaponized systems more responsive; these risks are not unique to our work but can be amplified by faster, more frequent replanning. We therefore recommend deployment practices that combine application-level access control, safety constraints with monitoring, and domain-appropriate human oversight, especially in high-stakes or human-facing settings. Our experiments are conducted in simulated embodied environments and on a single-arm laboratory robot, and do not involve human subjects or the collection of personal data.

## Acknowledgments

This work was supported by the Guangdong Provincial Key Lab of Integrated Communication, Sensing and Computation for Ubiquitous Internet of Things (No. 2023B1212010007).

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

## A. Appendix Details

This appendix provides additional material that supports the paper's core claims about (i) cost-aware token pruning (E-RECAP) as a reusable efficiency primitive and (ii) BRACE as a stability-aware budgeting controller with auditable replanning costs.

### A.1. E-RECAP Details

Figure 9 summarizes the E-RECAP experimental setup and where the pruning module is integrated on the replanning call path. We summarize key E-RECAP formulas and configuration details used throughout the paper. Given a training loss $\ell(X)$ and token representation $X_i$, the gradient-based saliency score is:

$$\hat{\pi}_i = |\langle \nabla_{X_i} \ell(X), X_i \rangle|, \quad i \in \{1, \ldots, N\}. \quad (7)$$

At inference, E-RECAP uses a lightweight predictor to estimate token importance from hidden states. Let $h_i \in \mathbb{R}^d$

*Table 8.* E-RECAP pruning module configuration. The table lists the hidden-state source, token scoring rule, protected head/tail windows, pruning layers, and keep-ratio settings used by the module.

| Parameter | Value |
|---|---|
| Pruning layers $\mathcal{P}$ (28-layer) | $\{4, 7, 10, 13, 16, 19, 22, 25\}$ |
| Pruning layers $\mathcal{P}$ (32-layer) | $\{4, 7, 10, 13, 16, 19, 22, 25, 28\}$ |
| Head/Tail preserved | $4 / \max(16, \lceil 0.1 N_l \rceil)$ |
| Keep ratios $r$ (per pruning layer) | $\{0.9, 0.8, 0.7, 0.6, 0.5\}$ |
| Intermediate dim. | $d/4$ |

*Table 9.* RoboFactory Pass-Shoe under the harder setting. In contrast to the saturated regime reported in the original variants table, this configuration exposes the necessity of replanning together with controller-side stabilization.

| Method | Success | Wait | Lat P95 | SLO viol. | Replans/Ep |
|---|---|---|---|---|---|
| Open-loop | 0.0% | 0.00 | – | – | 0.00 |
| Frozen plan | 0.0% | 55.64 | 72.78 | 0.0% | 1.00 |
| No BRACE | 0.0% | 6607.30 | 312.65 | 27.6% | 30.40 |
| **BRACE + E-RECAP** | **80.0%** | **6241.74** | **247.21** | **4.6%** | 36.80 |

denote the hidden representation for token $i$. The predicted importance score is:

$$\pi_i = W_2 \, \sigma(W_1 h_i + b_1) + b_2, \quad i \in \{1, \ldots, N\}, \quad (8)$$

where $\sigma$ is a nonlinearity (e.g., GELU) and the intermediate width is set to $d/4$.

We train the predictor with a combined objective:

$$\mathcal{L} = \mathcal{L}_L + \lambda_1 \mathcal{L}_M + \lambda_2 \mathcal{L}_R, \quad (9)$$

where $\mathcal{L}_L$ preserves generation quality, $\mathcal{L}_M$ matches saliency magnitudes, and $\mathcal{L}_R$ preserves relative ordering. A pairwise ranking term can be written as:

$$\mathcal{L}_R = \sum_{i=1}^{N-1} \sum_{j=i+1}^{N} \log\Big(1 + \exp\big(-(\pi_i - \pi_j)\cdot \\ \text{sign}(\hat{\pi}_i - \hat{\pi}_j)\big)\Big). \quad (10)$$

Finally, E-RECAP prunes progressively across a set of intermediate layers $\mathcal{P}$: at a pruning layer $l \in \mathcal{P}$ with sequence length $N_l$, it keeps $N_{l+1} = \lfloor r N_l \rfloor$ tokens (always preserving a small set of head/tail tokens), and selects the remaining tokens by top-$k$ importance scores. The layer schedules and keep ratios used in this paper are listed in Appendix Table 8. Because pruning is progressive, the overall token reduction reported in the experiments is measured empirically and is not simply $1 - r$.

**Pruning configuration.** Table 9 isolates the harder Pass-Shoe setting where recovery and stabilization are necessary for task completion.

**Per-domain setup parameters.** Table 11 summarizes the per-domain evaluation setup used throughout the paper. Token budgets apply to tokens passed into the replanning call

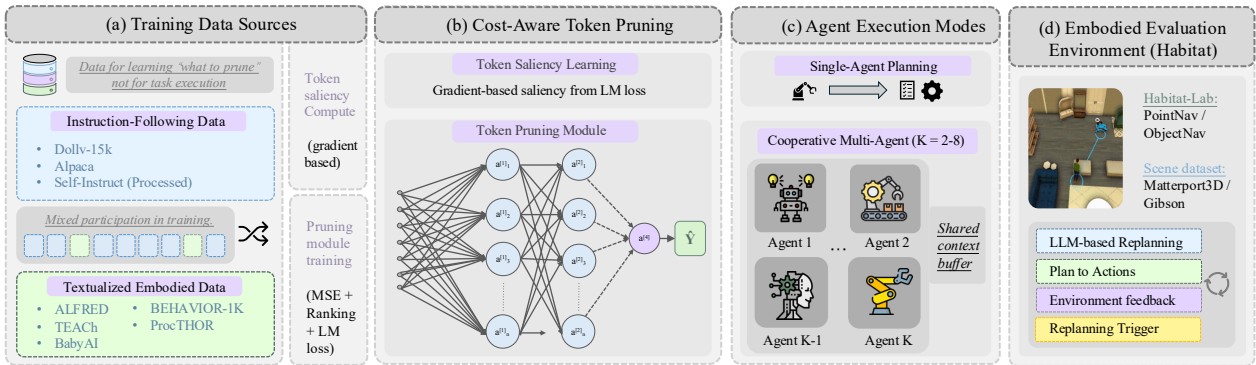

*Figure 9.* E-RECAP experimental setup overview. The diagram summarizes the replanning data flow, pruning module, and planner interface used for the Habitat acceleration experiments.

*Table 10.* Suppressed-trigger outcome audit. We distinguish suppressed triggers that are later recovered, harmless, or harmful within a short horizon, rather than treating all suppressed events as equivalent.

| Gate reason | Segments | Recovered | Harmless | Harmful | Override within H |
|---|---|---|---|---|---|
| Commit window | 13 | 13 (100.0%) | 0 (0.0%) | 0 (0.0%) | 13 (100.0%) |
| Cooldown | 34 | 23 (67.6%) | 9 (26.5%) | 2 (5.9%) | 25 (73.5%) |

*Table 11.* Per-domain setup parameters. The table reports episode counts, SLOs, agent counts, and token-budget settings for the main experimental domains and appendix variants.

| Platform | Ep/var | SLO (ms) | Agents | Budget(s) |
|---|---|---|---|---|
| Meta Habitat | 30 | 2,500 | 1 | $B=20$ (budget-match) |
| RoboFactory | 10 | 250 | multi-agent | $B=128$ (RAG×Prune) |
| Microsoft AirSim | 10 | 2,500 | $K=8$ | N/A |

after any compression/budgeting. Notes: Meta Habitat uses navigation with shortest-path-noise execution; RoboFactory reports coordination wait in addition to SLO/tail latency; AirSim reports safety proxies (near-miss/collision and minimum distance) alongside replanning cost.

**Additional E-RECAP ObjectNav results.** The main paper reports the PointNav rows in Table 5. Table 13 provides the corresponding ObjectNav extension and shows that the same pruning regime remains effective beyond a single navigation task family.

### A.2. Additional Domain Tables

The following tables provide additional per-domain results that complement the main-paper evidence snapshot by exposing stability and safety proxy metrics that are easy to hide when reporting only success or average latency (Qin et al., 2025; Shah et al., 2018; Sermanet et al., 2025).

**Clarification overhead (Habitat).** Table 14 reports a small clarification axis on Habitat navigation under the same replanning latency SLO used in the main paper. Clarification

adds non-trivial latency on the replanning call path and does not eliminate deadline-miss behavior in this setting, motivating the paper's focus on budgeting and tail-aware reporting. All settings here achieve 100% success, so we omit success-based conclusions and focus on tail latency and SLO violations.

**Meta Habitat variants.** Table 15 reports the full Meta Habitat variant table used by the motivating example in the main paper. Why it matters: under a matched token budget, different context-reduction strategies can yield materially different tail/SLO behavior, so we report tail percentiles and violation rates rather than success alone. All variants achieve 100% success in this setting, so we omit the Success column and focus on SPL and replanning cost/stability metrics. E-RECAP variants use keep ratio $r = 0.7$; budget-matched truncations enforce $B = 20$.

**Habitat phase overhead.** Table 16 separates E-RECAP-side overhead from planner latency on the Habitat slice. The key point is that pruning overhead remains small relative to planner time, so the main system gain comes from shortening the planner input under the same call-path accounting.

**Habitat direct replanning effect.** Table 17 makes explicit what changes when repeated replanning is disabled and when the same repeated-replanning regime is budgeted. On this easy PointNav slice, the principal difference is schedulability rather than task-quality collapse.

**RoboFactory variants.** Table 18 reports the full RoboFactory Pass-Shoe variant table. Why it matters: success saturates in this setting, so tail latency and SLO violations expose stability gaps that are invisible under success-only reporting. Tokens are mean tokens passed into replanning; Lat P95 is replanning latency (ms). All variants achieve 100% success, so we omit the Success column.

*Table 12.* Simulation platform, scene, and task coverage across the broader evaluation suite.

| Category | Platform | Scene family | Reported task / setting | Role | Status | Where used |
|---|---|---|---|---|---|---|
| Navigation | Meta Habitat | MP3D | PointNav | core navigation benchmark | reported | main text + appendix |
| Navigation | Meta Habitat | MP3D + shortest-path noise | PointNav with shortest-path noise | stress-test slice | reported | main text + appendix |
| Manipulation | RoboFactory | indoor handoff | Pass-Shoe | core manipulation benchmark | reported | main text + appendix |
| Manipulation | RoboFactory | harder handoff | Pass-Shoe harder-setting | harder-setting manipulation slice | reported | main text + appendix |
| Manipulation | RoboFactory | multi-agent photo task | TakePhoto | qualitative manipulation example | reported | appendix figure |
| Manipulation | RoboFactory | camera alignment | CameraAlignment | additional manipulation coverage | reported | appendix coverage |
| Manipulation | RoboSuite | dual-arm assembly | Two-Arm Peg-in-Hole | failure-case extension | reported | appendix figure |
| Manipulation | RoboSuite | dual-arm transfer | Two-Arm Handover | success-case extension | reported | appendix figure |
| Manipulation | LIBERO | kitchen scene 3 | Moka Pot task | cross-benchmark qualitative extension | reported | appendix figure |
| Manipulation | RoboCasa | kitchen / fruit-style tasks | PickPlaceCounterToCabinet, fruit-style tasks | additional task-family coverage | partial | coverage only |
| Traffic/UAV | Microsoft AirSim | AirSimNH | intersection / crossing | core traffic/UAV benchmark | reported | main text + appendix |
| Traffic/UAV | Microsoft AirSim | AbandonedPark | delayed replanning realism bundle | additional AirSim coverage | reported | appendix coverage |
| Traffic/UAV | Microsoft AirSim | LandscapeMountains | baseline vs BRACE side-by-side ($K=8$) | terrain-shift qualitative example | reported | appendix figure |
| Traffic/UAV | Isaac Sim | disturbance replay / parking | delayed replanning numeric evidence | additional simulator coverage | partial | appendix coverage |

*Table 13.* E-RECAP replanning acceleration on Habitat-Lab (MP3D) ObjectNav at $r = 0.7$. The rows follow the same layout as the PointNav comparison and report success, SPL, token count, latency, speedup, and token reduction across agent counts.

| $K$ | Method | Keep Ratio | Success | SPL | Tokens/Replan | Latency (s) | Speedup | Token Reduction |
|---|---|---|---|---|---|---|---|---|
| | No-Pruning | 1.0 | 0.82 | 0.68 | 3,156 | 2.67 | 1.00× | 0% |
| 1 | Random | 0.7 | 0.74 | 0.61 | 912 | 1.28 | 2.09× | 71.1% |
| | **E-RECAP** | **0.7** | **0.81** | **0.67** | **912** | **1.28** | **2.09×** | **71.1%** |
| | No-Pruning | 1.0 | 0.81 | 0.67 | 13,892 | 10.45 | 1.00× | 0% |
| 4 | Random | 0.7 | 0.67 | 0.58 | 3,701 | 4.48 | 2.33× | 73.4% |
| | **E-RECAP** | **0.7** | **0.80** | **0.66** | **3,701** | **4.48** | **2.33×** | **73.4%** |
| | No-Pruning | 1.0 | 0.77 | 0.64 | 42,156 | 31.23 | 1.00× | 0% |
| 8 | Random | 0.7 | 0.62 | 0.55 | 9,987 | 11.84 | 2.64× | 76.3% |
| | **E-RECAP** | **0.7** | **0.76** | **0.63** | **9,987** | **11.84** | **2.64×** | **76.3%** |

*Table 14.* Habitat clarification axis (10 episodes; replanning SLO=2500 ms).

| Setting | Turns | Clarif tok | Clarif lat (ms) | Lat P95 | SLO viol (%) |
|---|---|---|---|---|---|
| Coarsened goal (0 turn) | 0 | 0.0 | 0.0 | 2,653 | 76.3 |
| Coarsened goal (2 turn) | 2 | 54.5 | 845.0 | 2,668 | 100.0 |
| Coarsened process (2 turn) | 2 | 37.0 | 670.0 | 2,668 | 52.6 |
| Coarsened success (2 turn) | 2 | 39.0 | 690.0 | 2,671 | 89.5 |
| Oracle goal (0 turn) | 0 | 0.0 | 0.0 | 2,657 | 87.2 |

*Table 15.* Meta Habitat variants (30 episodes; latencies in ms). A no-initial-plan / no-replanning row is included to distinguish No BRACE from the open-loop condition. Shaded rows highlight No BRACE (blue) and BRACE(+E-RECAP) (yellow).

| Variant | SPL | Tokens | Lat P95 | SLO viol (%) | Token red (%) |
|---|---|---|---|---|---|
| No-initial-plan / no-replanning | 0.519 | – | – | N/A | – |
| No BRACE (baseline) | 0.991 | 235 | 2,677 | 85.5 | 0.0 |
| No BRACE + E-RECAP | 0.994 | **20** | **2,499** | **3.6** | 91.8 |
| No BRACE + Random budget-match | 0.990 | 20 | 2,605 | 33.3 | 91.7 |
| No BRACE + Recency budget-match | 0.994 | 20 | **2,302** | **0.0** | 92.2 |
| No BRACE + Structured summary | 0.987 | 20 | 2,605 | 36.4 | 92.0 |
| BRACE (no pruning) | 0.994 | 235 | 2,679 | 85.3 | 0.0 |
| **BRACE + E-RECAP** | 0.994 | **20** | 2,500 | 4.7 | 91.7 |

*Table 16.* Habitat phase-level overhead on the PointNav LLM-executor configuration. The explicit summarization stage measures 0.00 ms for all reported variants and is therefore omitted from the table.

| Variant | Prune mean | Planner mean | End-to-end mean | End-to-end P95 | Tok. after |
|---|---|---|---|---|---|
| No BRACE | 0.76 | 3804.74 | 3808.17 | 5492.08 | 256.85 |
| Pruning only (no BRACE gate) | 37.21 | 2414.07 | 2451.51 | 2488.24 | 20.21 |
| BRACE + E-RECAP | 35.63 | 2414.05 | 2449.92 | 2486.31 | 20.21 |
| Recency truncation B20 | 0.69 | 2287.56 | 2288.44 | 2289.53 | 20.00 |

inside one end-to-end latency number.

**RoboFactory budget-matched baselines** ($B = 128$). Table 21 keeps the original compact heuristic-only budget-matched table from the main submission. We preserve these previously reported values unchanged and then place the additional method expansion in a separate table.

Table 22 appends the additional method expansion on the same shared Pass-Shoe anchor. We keep these rows separate so that the new selector/recent-method comparison does not overwrite the original heuristic table.

**RoboFactory ambiguity $\times$ limited clarification.** Table 23 reports RoboFactory results under different ambiguity types and limited clarification turns. We report replanning tail latency and SLO violations to reflect deadline-miss regimes. Columns report clarification latency, replanning tail latency, and a coordination/execution wait-time proxy (all in ms), plus SLO violation rate (%).

**Microsoft AirSim variants.** Table 24 reports AirSim safety proxies alongside replanning cost metrics. Why it matters: in multi-agent interaction, safety proxies and deadline misses can diverge from success, so we expose both along with tail latency and SLO violations. Near-misses and collisions are per-episode safety proxies. All variants achieve 100% success, so we omit the Success column.

**AirSim replanning-frequency sweep (trigger audit).** Table 25 reports a trigger-audit summary for AirSim under a replanning-frequency sweep. We report effective replanning rates, how many triggers are suppressed by the controller,

**RoboFactory stability (coordination).** Table 19 reports coordination stability metrics (SLO violations and wait time). Wait is mean time spent waiting for coordination/execution (ms).

**Composable modules: retrieval and token pruning.** For completeness, we keep the original RoboFactory RAG×pruning ablation in the appendix after moving the stronger open-loop / harder-setting evidence to the main text. The point remains unchanged: retrieval overhead must be measured on the replanning call path rather than hidden

*Table 17.* Pairwise comparison on Habitat between the no-replanning boundary and budgeted repeated replanning. '–' denotes metrics not included for a given pair. On this PointNav configuration, the principal difference lies in schedulability rather than in task-quality degradation.

| Comparison | Task quality | Replans/ep | Wall ms | SLO % | Lat P95 (ms) |
|---|---|---|---|---|---|
| No-initial-plan → No BRACE | 53.3 / 0.519 → 53.3 / 0.515 | 0.000 → 4.533 | 94 → 17451 | – | – |
| No BRACE → BRACE + E-RECAP | 53.3 / 0.515 → 53.3 / 0.519 | – | – | 93.4 → 0.0 | 5492 → 2486 |
| No BRACE → Recency B20 | 53.3 / 0.515 → 53.3 / 0.519 | – | – | 93.4 → 0.0 | – |

*Table 18.* RoboFactory Pass-Shoe variants (10 episodes; latency SLO=250 ms). Shaded rows highlight baseline (blue) and BRACE(+E-RECAP) (yellow).

| Variant | Tokens | Lat P95 | SLO viol (%) | Token red (%) |
|---|---|---|---|---|
| No BRACE | 1,566 | 1,604 | 100.0 | 0.0 |
| No BRACE + E-RECAP | 350 | 1,236 | 100.0 | 77.6 |
| No BRACE + Recency baseline | 350 | 1,235 | 100.0 | 77.7 |
| BRACE | 1,414 | 1,587 | **50.0** | 0.0 |
| **BRACE + E-RECAP** | **319** | **1,213** | **50.0** | 77.4 |

*Table 19.* RoboFactory stability metrics (10 episodes; latency SLO=250 ms). Shaded rows highlight baseline (blue) and BRACE(+E-RECAP) (yellow).

| Variant | Tokens | Lat P95 (ms) | SLO viol (%) | Wait (ms) |
|---|---|---|---|---|
| No BRACE | 1,566 | 1,604 | 100.0 | 9,063 |
| BRACE | 1,414 | 1,587 | **50.0** | 4,213 |
| **BRACE + E-RECAP** | **319** | **1,213** | **50.0** | **3,546** |

*Table 20.* Joint ablation over RAG and pruning on RoboFactory (10 episodes; token budget $B = 128$). Token counts are reported after pruning and budgeting.

| RAG | Prune | Success (%) | Tokens | Retrieved tok | Retrieval (ms) | Lat P95 (ms) | SLO viol (%) |
|---|---|---|---|---|---|---|---|
| × | × | 47.6 | 208 | 0 | 0.0 | 326 | 30.6 |
| × | ✓ | 100.0 | **127** | 0 | 0.0 | **199** | **0.0** |
| ✓ | × | 0.0 | 237 | 28 | 37.9 | 393 | 68.3 |
| ✓ | ✓ | 100.0 | 128 | 28 | 37.9 | 240 | 2.3 |

and the trigger-type composition (Unsafe/Periodic/Failure as percentages of effective triggers). We report these trigger statistics for auditability under replanning pressure, not as a standalone outcome metric.

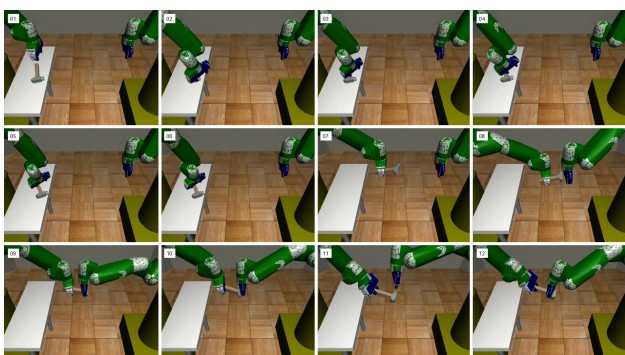

*Figure 10.* Qualitative extension on RoboSuite Two-Arm Handover. The contact sheet shows a successful coordinated manipulation rollout beyond the RoboFactory benchmark.

*Table 21.* Budget-matched baselines on RoboFactory ($B = 128$). Tail latency is the replanning latency at P95/P99 (ms), and the bind rate is reported as a percentage.

| Method | Tokens | Lat P95 | Lat P99 | Bind rate (%) |
|---|---|---|---|---|
| Random truncation | 128 | **200.8** | 239.6 | 94.9 |
| Recency truncation | 128 | 204.9 | **225.6** | **95.0** |
| Structured summary | 128 | 314.4 | 347.3 | 94.2 |

*Table 22.* Shared-anchor extension on RoboFactory Pass-Shoe ($B = 128$), reporting the additional methods evaluated under the same accounting contract.

| Method | Venue/Year | Success | Lat P95 | SLO viol. | Tok after | Wait | Type |
|---|---|---|---|---|---|---|---|
| **E-RECAP** | – | 100.0% | 207.09 | 0.49% | **125.66** | **5776.50** | learned pruning |
| Grad-Hidden Saliency | – | 100.0% | 239.67 | 2.43% | 125.98 | 7079.05 | derived selector |
| RAG Memory + E-RECAP | – | 100.0% | 241.52 | 3.47% | 126.61 | 7193.86 | retrieval + selector |
| RAG Static + E-RECAP | – | 100.0% | 241.61 | 3.48% | 126.61 | 7178.41 | retrieval + selector |
| TurboQuant | ICLR 2026 | 100.0% | 221.57 | 1.73% | 125.90 | 6021.51 | quantization |
| ReST-KV | ICLR 2026 | 100.0% | 231.95 | 2.27% | 125.81 | 6807.52 | selector |
| FreeKV | ICLR 2026 | 100.0% | 239.99 | 3.47% | 126.64 | 7269.27 | retrieval |
| DefensiveKV | ICLR 2026 | 100.0% | 239.09 | 3.49% | 125.83 | 6862.33 | selector |
| kvtc | ICLR 2026 | 100.0% | 210.33 | 1.01% | 125.86 | 5979.35 | quantization |

*Table 23.* Joint sweep over ambiguity type and the number of clarification turns on RoboFactory. Shaded rows denote the no-clarification setting (Turns=0).

| Ambiguity | Turns | Clarif lat | Tokens | Lat P95 | SLO viol (%) | Wait P95 |
|---|---|---|---|---|---|---|
| goal | 0 | 0 | 181 | 291 | 12.8 | 8,051 |
| goal | 1 | 87 | 207 | 330 | 29.5 | 9,189 |
| goal | 2 | 102 | 207 | 330 | 28.3 | 9,163 |
| process | 0 | 0 | 184 | 292 | 17.6 | 8,261 |
| process | 1 | 87 | 207 | 313 | 27.1 | 8,975 |
| process | 2 | 104 | 207 | 322 | 30.2 | 9,277 |
| success | 0 | 0 | 187 | 307 | 21.3 | 8,594 |
| success | 1 | 88 | 207 | 335 | 35.4 | 9,031 |
| success | 2 | 103 | 207 | 327 | 23.2 | 9,237 |

*Table 24.* Microsoft AirSim intersection variants ($K = 8$; 10 episodes; SLO=2500 ms).

| Variant | Near-miss/ep | Collisions/ep | Tokens | Tok red (%) | Lat P50 | Lat P95 | SLO viol (%) |
|---|---|---|---|---|---|---|---|
| No BRACE | 24.7 | 0.0 | 2,934 | 0.0 | 5,960 | 8,520 | 100.0 |
| **BRACE + E-RECAP** | 29.2 | 0.0 | **1,114** | **65.0** | **1,640** | **1,640** | **4.7** |

*Table 25.* AirSim trigger audit under a replanning-frequency sweep. The Interval column is measured in controller steps; percentages are computed relative to the effective triggers. Shaded rows highlight baseline (blue) and BRACE (yellow).

| Freq | Interval | Variant | Replans/min | Suppressed/ep | Triggers | Unsafe (%) | Periodic (%) | Failure (%) |
|---|---|---|---|---|---|---|---|---|
| 3.3× | 6 | No BRACE | 61.2 | 22.7 | 123 | 53.7 | 44.7 | 1.6 |
| 3.3× | 6 | **BRACE + E-RECAP** | 50.5 | 10.7 | 106 | 31.1 | 68.9 | 0.0 |

*Table 26.* AirSim comparison across agent counts $K$ ($K$ denotes the number of agents; latencies in ms). Shaded rows highlight baseline (blue) and BRACE (yellow).

| $K$ | Variant | Tokens | Tok red (%) | Lat P95 | Lat P99 | Min dist (m) |
|---|---|---|---|---|---|---|
| 1 | baseline | 3,550 | 0.0 | 12,684 | 13,177 | 16.79 |
| 1 | **BRACE** | **765** | **78.5** | **1,640** | **1,640** | 14.31 |
| 2 | baseline | 2,780 | 0.0 | 9,488 | 9,834 | 4.12 |
| 2 | **BRACE** | **770** | **75.7** | **1,640** | **1,640** | 4.02 |
| 4 | baseline | 4,520 | 0.0 | 15,560 | 16,136 | 8.05 |
| 4 | **BRACE** | **800** | **82.0** | **1,640** | **1,640** | 7.58 |

*Table 27.* E-RECAP backbone robustness (PointNav, MP3D; $K = 4$; $r = 0.7$; 200 episodes).

| Model | Layers | Method | Success | SPL | Tokens/Replan | Latency (s) | Speedup | Token Reduction |
|---|---|---|---|---|---|---|---|---|
| Qwen2-7B-Instruct | 28 | No-Pruning | 0.84 | 0.71 | 12,847 | 9.87 | 1.00× | 0% |
| | | **E-RECAP** | **0.83** | **0.70** | **3,421** | **4.23** | **2.33×** | **73.4%** |
| LLaMA-3-8B-Instruct | 32 | No-Pruning | 0.82 | 0.69 | 12,156 | 9.23 | 1.00× | 0% |
| | | **E-RECAP** | **0.81** | **0.68** | **3,306** | **4.05** | **2.28×** | **72.8%** |
| Mistral-7B-Instruct | 32 | No-Pruning | 0.85 | 0.72 | 11,892 | 9.45 | 1.00× | 0% |
| | | **E-RECAP** | **0.84** | **0.71** | **3,199** | **4.09** | **2.31×** | **73.1%** |
| Qwen2.5-7B-Instruct | 28 | No-Pruning | 0.84 | 0.71 | 12,234 | 9.67 | 1.00× | 0% |
| | | **E-RECAP** | **0.83** | **0.70** | **3,218** | **4.12** | **2.35×** | **73.7%** |
| LLaMA-2-7B-Chat | 32 | No-Pruning | 0.81 | 0.68 | 12,456 | 10.12 | 1.00× | 0% |
| | | **E-RECAP** | **0.80** | **0.67** | **3,425** | **4.48** | **2.26×** | **72.5%** |
| ChatGLM3-6B | 28 | No-Pruning | 0.79 | 0.66 | 11,234 | 8.89 | 1.00× | 0% |
| | | **E-RECAP** | **0.78** | **0.65** | **2,898** | **3.67** | **2.42×** | **74.2%** |

**AirSim showcase across different $K$.** Table 26 summarizes a small AirSim showcase across multiple agent counts $K$, using the same metrics as the main paper where applicable (tokens after compression and tail latency percentiles). We omit SLO-based columns when the underlying summaries do not define an SLO. All variants achieve 100% success in this showcase, so we omit the Success column.

**Additional E-RECAP results (Habitat-Lab).** This section reports additional E-RECAP-only results on Habitat-Lab navigation (MP3D) (Chang et al., 2017) that support the standalone pruning module evidence used throughout the BRACE paper. Training-data sources referenced in Table 28 include Dolly, Alpaca, and Self-Instruct (Conover et al., 2023; Taori et al., 2023; Wang et al., 2023). Figure 18 complements the tables by summarizing the compression–quality/efficiency tradeoff across pruning strengths. In Tables 27–28, $r$ denotes the per-pruning-layer keep ratio (Appendix Table 8); overall Token Reduction is measured empirically from Tokens/Replan relative to No-Pruning and is not constrained to $1 - r$. Speedup and Token Reduction are computed relative to No-Pruning under the same task and $K$ (↑ higher is better).

We keep an additional RoboFactory sanity check (Figure 11) to illustrate that the same qualitative expectation holds in a manipulation/coordination setting: E-RECAP should reduce replanning overhead without introducing obvious behavior drift. In this representative snapshot at the same nominal moment, BRACE+E-RECAP has already picked up and placed the shoe, while the baseline is still in the pickup stage, qualitatively reflecting faster task progress.

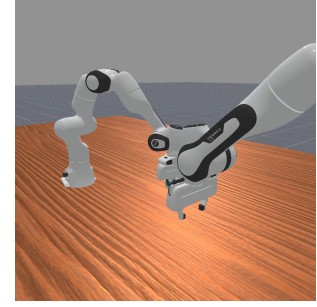 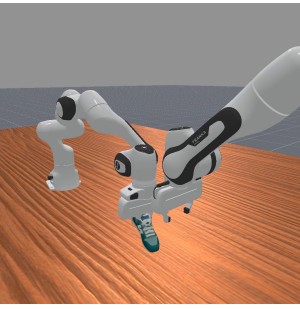

*(a)* Baseline.      *(b)* BRACE+E-RECAP.

*Figure 11.* Qualitative comparison on RoboFactory Pass-Shoe between the baseline and BRACE+E-RECAP. The paired frames illustrate how BRACE stabilizes replanning and reduces coordination delay during the manipulation handoff.

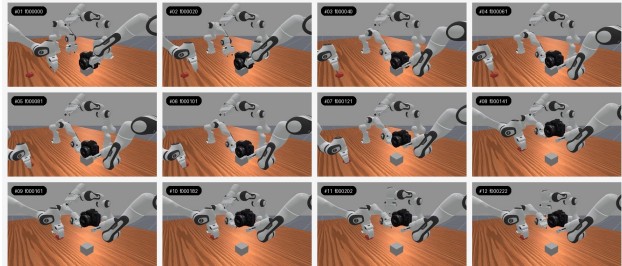

*Figure 12.* Additional qualitative example on RoboFactory TakePhoto. The frame sequence depicts a successful multi-agent manipulation rollout that complements the Pass-Shoe harder-setting results.

Main-paper Table 5 reports E-RECAP's replanning acceleration on Habitat-Lab (MP3D) under multi-agent context growth; below we provide additional supporting sweeps and configuration results. Figure 14, Table 28, Table 29, and Figure 18 summarize the supporting data-mixture, heuristic-baseline, and pruning-strength analyses.

## B. Controller Sweep (Proxy Environment)

This appendix section provides a controlled sweep that isolates controller stability mechanisms (cooldown and commit windows, plus a deadlock window parameter) on a proxy environment where success is sensitive to replanning churn and deadlocks.

In complex embodied benchmarks, deadlocks and thrashing can be confounded by perception noise, action failures, and task diversity. We therefore use a proxy setting that isolates controller dynamics: success depends primarily on whether replanning is executed stably (without excessive plan churn) and whether deadlocks are detected and resolved.

We vary three controller knobs: cooldown $\delta$ (minimum steps between replanning calls), commit window $\omega$ (minimum steps to execute a plan before reconsideration), and dead-

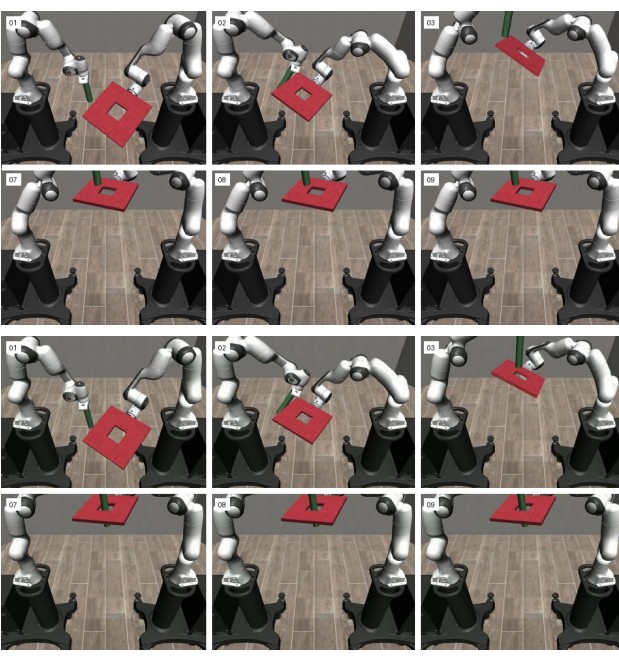

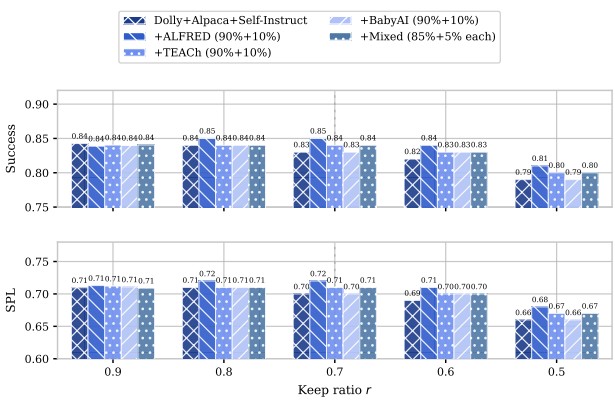

*Figure 13.* Full-rollout comparison on RoboSuite TWO-ARM PEG-IN-HOLE. The combined contact sheet contrasts a BRACE rollout that stabilizes replanning and successfully completes the task against the corresponding No-BRACE baseline, which repeatedly triggers replanning without controller-level budgeting or stabilization and ultimately fails.

*Figure 14.* Training-data configuration overview for E-RECAP. The figure summarizes the data mixture used by the token-utility predictor and corresponds to the quantitative comparison in Table 28.

*Table 28.* Training-data configuration effect for E-RECAP (Point-Nav, MP3D; $K = 4$; 200 episodes).

| Training Data Config | Keep Ratio | Success | SPL | Token Reduction | Latency (s) | Speedup | Quality Gain |
|---|---|---|---|---|---|---|---|
| *Instruction-Only (Baseline)* | | | | | | | |
| | 0.9 | 0.84 | 0.71 | 57.0% | 7.56 | 1.31× | baseline |
| | 0.8 | 0.84 | 0.71 | 65.2% | 5.64 | 1.75× | baseline |
| Dolly+Alpaca+Self-Instruct | **0.7** | **0.83** | **0.70** | **73.4%** | **4.23** | **2.33×** | **baseline** |
| | 0.6 | 0.82 | 0.69 | 79.2% | 3.52 | 2.80× | baseline |
| | 0.5 | 0.79 | 0.66 | 85.0% | 2.95 | 3.35× | baseline |
| *Instruction + Embodied Auxiliary Data* | | | | | | | |
| | 0.9 | 0.84 | 0.71 | 57.0% | 7.56 | 1.31× | +0.0% |
| | 0.8 | 0.85 | 0.72 | 65.2% | 5.64 | 1.75× | +1.2% |
| +ALFRED (90%+10%) | **0.7** | **0.85** | **0.72** | **73.4%** | **4.23** | **2.33×** | **+2.4%** |
| | 0.6 | 0.84 | 0.71 | 79.2% | 3.52 | 2.80× | +2.4% |
| | 0.5 | 0.81 | 0.68 | 85.0% | 2.95 | 3.35× | +2.5% |
| | 0.9 | 0.84 | 0.71 | 57.0% | 7.56 | 1.31× | +0.0% |
| | 0.8 | 0.84 | 0.71 | 65.2% | 5.64 | 1.75× | +0.0% |
| +TEACh (90%+10%) | **0.7** | **0.84** | **0.71** | **73.4%** | **4.23** | **2.33×** | **+1.2%** |
| | 0.6 | 0.83 | 0.70 | 79.2% | 3.52 | 2.80× | +1.2% |
| | 0.5 | 0.80 | 0.67 | 85.0% | 2.95 | 3.35× | +1.3% |
| | 0.9 | 0.84 | 0.71 | 57.0% | 7.56 | 1.31× | +0.0% |
| | 0.8 | 0.84 | 0.71 | 65.2% | 5.64 | 1.75× | +0.0% |
| +BabyAI (90%+10%) | **0.7** | **0.83** | **0.70** | **73.4%** | **4.23** | **2.33×** | **+0.0%** |
| | 0.6 | 0.83 | 0.70 | 79.2% | 3.52 | 2.80× | +0.0% |
| | 0.5 | 0.79 | 0.66 | 85.0% | 2.95 | 3.35× | +0.0% |
| | 0.9 | 0.84 | 0.71 | 57.0% | 7.56 | 1.31× | +0.0% |
| | 0.8 | 0.84 | 0.71 | 65.2% | 5.64 | 1.75× | +0.0% |
| +Mixed (85%+5% each) | **0.7** | **0.84** | **0.71** | **73.4%** | **4.23** | **2.33×** | **+1.2%** |
| | 0.6 | 0.83 | 0.70 | 79.2% | 3.52 | 2.80× | +1.2% |
| | 0.5 | 0.80 | 0.67 | 85.0% | 2.95 | 3.35× | +1.3% |

*Table 29.* Heuristic baseline vs E-RECAP (PointNav, MP3D; $K = 4$; $r = 0.7$; 200 episodes).

| Method | Success | SPL | Tokens/Replan | Latency (s) | Speedup |
|---|---|---|---|---|---|
| No-Pruning | 0.84 | 0.71 | 12,847 | 9.87 | 1.00× |
| Random-Pruning | 0.65 | 0.57 | 3,421 | 4.23 | 2.33× |
| Heuristic-Pruning (recency) | 0.78 | 0.67 | 3,421 | 4.23 | 2.33× |
| **E-RECAP** | **0.83** | **0.70** | **3,421** | **4.23** | **2.33×** |

lock window $w$ (a short-horizon aggregation window for proxy deadlock signals). Intuitively, larger $\delta$ and $\omega$ reduce churn, while $w$ controls how quickly deadlock evidence accumulates to trigger corrective behavior.

We can summarize the proxy controller logic with a simple stability gate and deadlock evidence accumulator. Let $\tau_t$ denote a proxy trigger (e.g., periodic or deadlock recovery), let $\Delta_t$ and $\kappa_t$ denote the cooldown/commit counters, and let $z_t \in \{0, 1\}$ denote a per-step proxy deadlock indicator. We execute replanning when:

$$u_t = \mathbb{I}[\tau_t \wedge (\Delta_t \geq \delta) \wedge (\kappa_t \geq \omega)]. \quad (11)$$

Deadlock evidence is aggregated over a short horizon $w$:

$$Z_t^{(w)} = \sum_{i=t-w+1}^{t} z_i, \quad (12)$$

where larger $w$ makes recovery less reactive (but potentially less noisy). We report planner calls per episode, plan changes per episode, and define a churn ratio $\chi$ as:

$$\chi = \frac{\text{Changes/ep}}{\text{Calls/ep}}. \quad (13)$$

**Deterministic anti-churn property.** In the absence of failure-aware overrides, Eq. (11) immediately implies two

*Table 30.* Controller stability sweep providing the quantitative audit underlying the anti-churn discussion. In the absence of failure-aware overrides, larger cooldown and commit windows reduce plan churn but can also degrade task success when they become overly restrictive.

| Variant | $\delta$ | $\omega$ | $w$ | Success (%) | Calls/ep | Changes/ep | $\chi$ | Deadlocks/ep | Stall/ep |
|---|---|---|---|---|---|---|---|---|---|
| No BRACE + E-RECAP | N/A | N/A | N/A | 5.0 | 167.18 | 166.08 | 0.993 | 157.53 | 225.10 |
| **BRACE + E-RECAP** | 0 | 0 | 1 | 98.3 | 14.37 | 13.10 | 0.912 | 0.00 | 91.20 |
| BRACE + E-RECAP | 0 | 0 | 3 | 25.0 | 95.72 | 94.45 | 0.987 | 88.47 | 189.58 |
| BRACE + E-RECAP | 0 | 0 | 5 | 6.7 | 159.68 | 158.50 | 0.993 | 154.63 | 222.97 |
| BRACE + E-RECAP | 0 | 2 | 1 | 75.0 | 24.40 | 23.33 | 0.956 | 14.78 | 131.93 |
| BRACE + E-RECAP | 0 | 2 | 3 | 13.3 | 125.98 | 124.85 | 0.991 | 121.15 | 211.17 |
| BRACE + E-RECAP | 0 | 2 | 5 | 6.7 | 165.50 | 164.40 | 0.993 | 162.13 | 219.53 |
| BRACE + E-RECAP | 3 | 0 | 1 | 81.7 | 22.90 | 21.73 | 0.949 | 11.12 | 129.42 |
| BRACE + E-RECAP | 3 | 0 | 3 | 18.3 | 110.73 | 109.60 | 0.990 | 104.43 | 197.65 |
| BRACE + E-RECAP | 3 | 0 | 5 | 5.0 | 175.25 | 174.15 | 0.994 | 171.02 | 227.07 |
| BRACE + E-RECAP | 3 | 2 | 1 | 78.3 | 21.78 | 20.72 | 0.951 | 12.12 | 125.32 |
| BRACE + E-RECAP | 3 | 2 | 3 | 15.0 | 141.27 | 140.20 | 0.992 | 137.28 | 207.17 |
| BRACE + E-RECAP | 3 | 2 | 5 | 3.3 | 161.98 | 160.90 | 0.993 | 158.37 | 225.80 |

controller-side constraints: (i) if a replanning call is executed at step $t$, then another call cannot be admitted before the cooldown counter satisfies $\Delta_{t'} \geq \delta$, so consecutive calls are separated by at least $\delta$ controller steps; and (ii) if a plan update is accepted at step $t$, then another replacement cannot be admitted before the commit counter satisfies $\kappa_{t'} \geq \omega$, so the updated plan must survive at least $\omega$ controller steps before reconsideration. *Proof sketch:* both statements follow directly from the gate conjunction because, without overrides, violating either threshold forces $u_{t'} = 0$ until the corresponding counter reaches its threshold. This is a deterministic spacing property, not a theorem-level guarantee on the full embodied closed loop. Appendix Table 30 reports the sweep under a fixed token budget and keep ratio; proxy SLO violations are 0% for all variants. Figure 15 visualizes the same sweep to make the success/deadlock tradeoff easier to inspect.

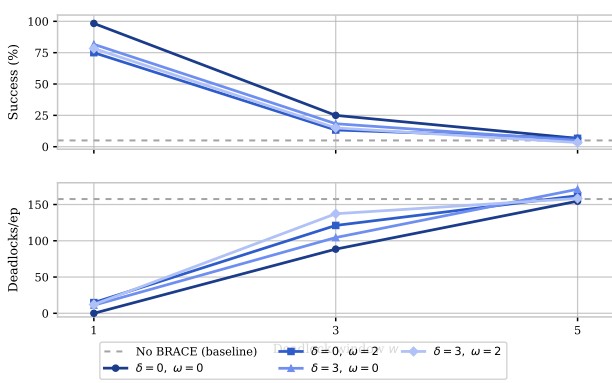

*Figure 15.* Visualization of the proxy controller sweep. Success and Deadlocks/ep are plotted as a function of the deadlock window $w$, with separate lines for different cooldown $\delta$ and commit window $\omega$ settings (Table 30); the dashed line marks the No-BRACE baseline.

Without BRACE, the system replans nearly every step, producing extreme churn and persistent deadlocks. With BRACE, modest stabilization (small $\delta$ and $\omega$) collapses planner calls and deadlocks, yielding high success. However,

overly large deadlock windows (larger $w$) degrade stability by delaying or mis-aggregating deadlock evidence, increasing both deadlocks and replanning churn in this proxy setting.

## C. Focused Real-Robot Evaluation

Figures 12, 13, 10, 16, and 17 provide additional qualitative rollouts used to audit behavior beyond the main RoboFactory and AirSim examples.

To complement the simulation-first evidence in the main paper, we include a focused single-arm real-robot evaluation on PICKFRUIT and PUSHT. The goal is not to claim broad deployment completeness, but to show that the same budgeting and replanning interface remains meaningful in physical execution under a fixed hardware/software stack. Table 31 documents the physical setup, and Table 32 gives the detailed per-task outcomes behind the compact main-text summary.

*Table 31.* Real-robot setup for the focused physical evaluation. The table lists the robot, camera stack, model inputs, planner and executor choices, task set, and episode budget.

| Component | Choice |
|---|---|
| Robot | Songling PiPER, single-arm |
| Robot control stack | `piper_sdk` over CAN |
| Cameras | 2 × Orbbec RGB-D + 1 record-only overview camera |
| Model input modality | RGB only |
| Camera SDK | OrbbecSDK v1 |
| LLM planner | Qwen2.5-14B-Instruct |
| VLA executor | OpenVLA |
| Real-robot tasks | PickFruit, PushT |
| Evaluation budget | 25 episodes / method / task / condition |
| Video categories | Motivation, Replanning in Action, Method Comparison |

*Table 32.* Detailed real-robot results for PICKFRUIT and PUSHT. The table expands the main-text summary with episode counts, success counts, duration, replanning frequency, tail replanning latency, SLO violations, and task-specific failure categories.

| Task | Method | Ep | Succ. | Rate | Dur. (s) | Replans/Ep | P95 Replan (s) | SLO Viol. | Failure A | Failure B |
|---|---|---|---|---|---|---|---|---|---|---|
| PICKFRUIT | One-Shot | 25 | 2/25 | 8.0% | 402 | 0.0 | N/A | N/A | grasp failure=11 | drop/slip=6 |
| PICKFRUIT | No BRACE | 25 | 6/25 | 24.0% | 458 | 3.4 | 29.4 | 42.7% | grasp failure=9 | drop/slip=5 |
| PICKFRUIT | **BRACE + E-RECAP** | 25 | 10/25 | **40.0%** | 369 | 1.9 | 22.8 | 18.6% | grasp failure=8 | drop/slip=4 |
| PUSHT | One-Shot | 25 | 0/25 | 0.0% | 579 | 0.0 | N/A | N/A | goal error>thr=14 | contact loss=12 |
| PUSHT | No BRACE | 25 | 3/25 | 12.0% | 632 | 3.8 | 34.7 | 61.3% | goal error>thr=12 | contact loss=15 |
| PUSHT | **BRACE + E-RECAP** | 25 | 8/25 | **32.0%** | 501 | 2.2 | 26.1 | 27.5% | goal error>thr=8 | contact loss=7 |

**Mechanism-facing real-robot evidence.** Beyond final success rates, we keep four concise mechanism-facing notes for the real-robot package.

- PICKFRUIT motivation subset: in the ambiguous-prompt subset ($n=10$), **BRACE + E-RECAP** triggers 2.4 replans per episode and 1.1 clarification requests per episode, versus 1.0 and 0.3 on the clear-prompt subset.

- PUSHT motivation subset: drift/contact-offset events trigger 2.0 replans per episode on average, and the

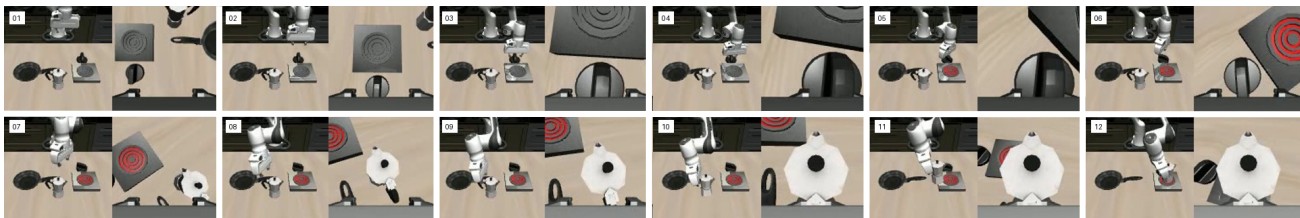

*Figure 16.* Successful BRACE rollout on the LIBERO Kitchen Scene 3 Moka Pot task. The replanning interface generalizes to a kitchen manipulation scenario with a different visual and action distribution.

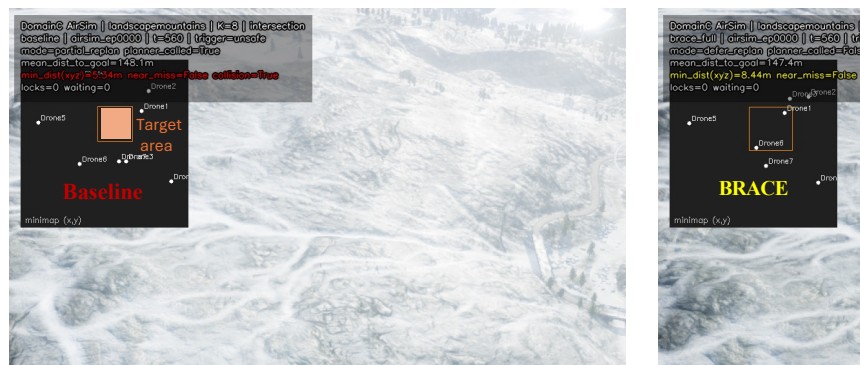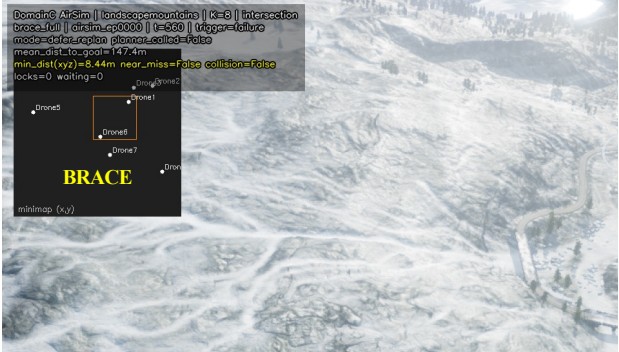

*Figure 17.* Qualitative example on the Microsoft AirSim LandscapeMountains scene ($K=8$ agents). Left: a baseline that continues to trigger replanning after collision- or disturbance-induced drift but does not apply BRACE's controller-level budgeting and stabilization. Right: BRACE, which decides whether to honor each trigger, allocates the per-call budget $B_t$ and latency target $SLO_t$ when replanning is admitted, and suppresses replanning churn via cooldown, commit, and failure-aware override. Frames are taken at the same time step and show one drone's first-person view together with the task-region third-person view; the orange overlay marks the goal region. Under BRACE the agents have already reached task completion, whereas the baseline rollout has not yet reached the goal region.

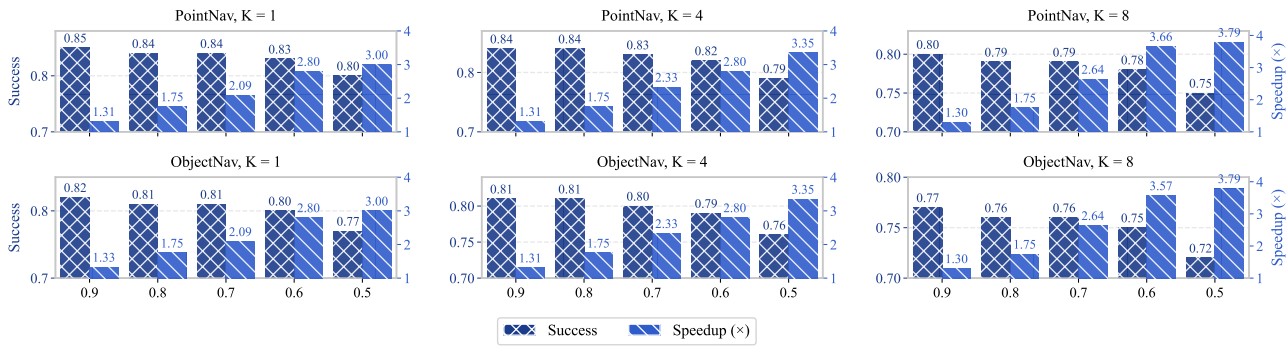

*Figure 18.* E-RECAP tradeoff between compression and performance under stronger pruning. The plot shows how latency and token savings change as the keep ratio decreases, alongside the resulting task-quality trend.

median goal error decreases by 22.1% within 13.2s after the trigger.

- PICKFRUIT trigger-centered rollout: in the banana example, the first grasp failure occurs at $t=173$s, the replanning update is issued at $t=186$s, and the robot completes successful grasp-and-place at $t=241$s.

- PUSHT trigger-centered rollout: in the physical rollout, contact loss occurs at $t=229$s, the replanning trigger is issued at $t=236$s, contact is recovered at $t=278$s, and the task completes at $t=431$s.

