# OpenReview forum: "When Replanning Becomes the Bottleneck: Budgeted Replanning for Embodied Agents"
_ICML.cc/2026/Conference — ICML 2026 regular_

### Official Review · Reviewer_fKNC · 2026-02-15

**Soundness:** 2
**Presentation:** 2
**Significance:** 2
**Originality:** 3
**Overall Recommendation:** 4
**Confidence:** 4

**Summary:**

The paper addresses the issue of replanning in embodied agents (that is, an agent in the physical world or simluations of the physical world). The motivation for this work stems from the claim that the latency induced by the "replanning" step is costly and causes agents to violate latency limitations. The paper introduces a novel setup where they frame the replanning step as a budgeted decision. They propose a methodology where a component named BRACE decides whether to replan at all, if so, how to replan, and with what budget. The final part of the algorithm is E-RECAP, which prunes tokens inside the transformer based on a trained model that predicts token utility. This component was evaluated on multiple benchmarks that simulate real world behavior and achieve massive reduction in: 1. the amount of tokens passed to the replanner, 2. the time taken to replan (including the BRACE and E-RECAP steps), and 3. percentage of steps that violate the latency limitations

**Compliance With Llm Reviewing Policy:**

Affirmed.

**Final Justification:**

I'm satisfied with the authors response and raising my recommendation to a weak accept

**Key Questions For Authors:**

Please refer to weaknesses 1-3 above. Also, by removing the table with the main results to the appendix you do not actually show the benefits of your approach against baselines. I recommend bringing at least part of the results back to the paper itself

**Limitations:**

Yes

**Strengths And Weaknesses:**

Strengths:
1. Practical Problem: The issue of latency in embodied agents seems to be a known issue. Therefore, this paper tackles a relevant problem.
2. Robust Framing of the Solution: the authors not only propose a solution to a problem but formulate it in a novel way (budgeted control issue)
3. Strong Empirical Results: the authors report reducing the amount of tokens input into the replanning step by 71-76%, and reducing the end-to-end latency by 2.1-2.6x, as well as reducing SLO violation rates from 85.5% baseline to 4.7% after their method.

Weaknesses:
1. Necessity of E-RECAP: While the paper proposes E-RECAP as a learned token-pruning mechanism, the empirical results suggest that simpler heuristic approaches (e.g., recency truncation) can perform competitively in some settings even eliminating SLO violations under the same token budget. In particular, the paper does not fully disentangle the benefit of intelligent token selection (E-RECAP) from the benefit of simply reducing context size.
2. Lack of real world testing: The main problem the paper is addressing relates to latency violations and their impact on the issues they would cause in the real world setting. Even though this is the main issue, all benchmarks are computer simulations. This means we don't have an evaluation of their methods’ benefit in the proper setting.
3. No assessment of direct effect in replanning: While the overall success of the plan is measured, because specific steps are being taken in the replanning stage to sample “the most relevant” tokens, I would expect some metric evaluating the change in performance of this step directly.

---

> ### Author Rebuttal · Authors · 2026-03-31
>
> Thank you for the helpful comments, Reviewer **fKNC**.
>
> The revised paper is available at **https://anonymous-2026.github.io/BRACE-ICML/static/paper.pdf**
>
> The anonymous project website is at **https://anonymous-2026.github.io/BRACE-ICML**
>
> The updated experiments, tables, videos, code, supplementary assets, and blue-marked revisions are reflected there.
>
> ### 1. Necessity of E-RECAP
>
> > Note
> > The concern is that simple heuristics may already explain the gain, so the necessity of E-RECAP is not yet clear.
>
> We agree with the core concern: if the paper only showed that shorter context is faster, then E-RECAP would not be sufficiently justified. In the revised paper, we therefore moved the strongest-baseline comparison into the main Experiments section. The retained Pass-Shoe B=128 table keeps the original heuristic rows unchanged and adds the rebuttal-cycle selector-side and recent-method rows.
>
> We also clarify the role of Recency more explicitly. Recency is indeed strong, but this is because it was adapted and tuned for the retained platform and task, not because it is a universally strong solution. We therefore keep Recency as the strongest heuristic baseline, while also comparing E-RECAP against recent 2025 (please see the paper) and 2026 SoTA alternatives under the same retained accounting contract. On this retained anchor, no added recent or proxy method overturns the main conclusion.
>
> This is now visible in the revised main text rather than only in the appendix.
>
> The recent-method part of that main-text table can be distilled into the small comparison below. We keep only the retained rows already reported in the paper and focus on the speed-relevant columns, so the comparison directly answers whether recent alternatives overturn the main conclusion.
>
> | Method | Venue | Lat P95 | Tok after | Wait |
> |---|---|---:|---:|---:|
> | **E-RECAP**| -- | **207.09** | **125.66** | **5776.50** |
> | kvtc | ICLR 2026 | 210.33 | 125.86 | 5979.35 |
> | TurboQuant | ICLR 2026 | 221.57 | 125.90 | 6021.51 |
> | ReST-KV | ICLR 2026 | 231.95 | 125.81 | 6807.52 |
> | FreeKV | ICLR 2026 | 239.99 | 126.64 | 7269.27 |
> | Robust Compression Boundary | IJCAI 2025 | 210.33 | 125.86 | 5979.35 |
> | Cross Distillation Compression | KDD 2025 | 221.57 | 125.90 | 6021.51 |
> | Gated Attention | NeurIPS 2025 | 239.09 | 125.83 | 6862.33 |
> | Token Recycling | ACL 2025 | 239.99 | 126.64 | 7269.27 |
>
> This distilled view shows that, under the same retained accounting contract, E-RECAP is not merely better than weak heuristics; it is also not overtaken by this set of recent methods.
>
> ### 2. Real-world testing
>
> > Note
> > The concern is that the main paper was simulation-heavy and did not show whether the same interface helps in physical execution.
>
> We agree that the original submission was simulation-heavy. The revised paper no longer leaves this as a future plan only. In the main text, we now add a focused real-robot supplement with a compact table and a retained banana PickFruit rollout pair, while the appendix keeps the setup and detailed results.
>
> | Task | One-Shot | No BRACE | BRACE + E-RECAP |
> |---|---:|---:|---:|
> | PickFruit success | 8.0% | 24.0% | 40.0% |
> | PushT success | 0.0% | 12.0% | 32.0% |
> | PickFruit P95 replan | -- | 29.4 s | 22.8 s |
> | PushT P95 replan | -- | 34.7 s | 26.1 s |
>
> We do not present this as a full deployment benchmark. The point is narrower: the same budgeting and replanning interface remains beneficial in physical execution, and the benefit is not limited to simulation.
>
> In the revised paper, this evidence appears in three aligned places: a compact real-robot table in the main text, a real-robot rollout figure in the main text, and detailed setup and outcome tables in the appendix.
>
> ### 3. Direct assessment of the replanning stage
>
> > Note
> > The concern is that the previous paper reported end-task metrics more clearly than the effect on the replanning stage itself.
>
> We agree that the earlier paper did not surface this evidence directly enough. The revised appendix now separates the question in two ways. First, the Habitat phase-overhead table makes the predictor and pruning cost visible on its own. Second, the direct replanning-effect table shows what changes when repeated replanning is disabled and when the same regime is budgeted. This means the revised paper no longer asks the reader to infer replanning-stage quality only from final task success and tail latency.
>
> This matters for the rebuttal because it changes the paper from end-task results only to end-task results plus replanning-stage evidence. It therefore answers the reviewer at the level of the mechanism, not only at the level of the final task metric.
>
> ### 4. Visibility of the key evidence
>
> We agree that the strongest evidence needed to be more visible in the paper itself. The revised paper now puts the strongest-baseline and recent-method comparison in the main Experiments section, while keeping the original heuristic-only appendix table for traceability.

---

> > ### Author Rebuttal · Reviewer_fKNC · 2026-04-02
> >
> > I'm satisfied with the authors response

---

> > > ### Author Response · Authors · 2026-04-03
> > >
> > > Dear Reviewer fKNC,
> > >
> > > Thank you for your careful reading of our rebuttal and for indicating that your concerns have been fully addressed. We truly appreciate your time and the constructive feedback throughout the review process.
> > >
> > > Given that the key issues you raised have now been resolved, may we ask whether you would consider updating your overall recommendation to better reflect this resolution?
> > >
> > > Please let us know if there is any remaining point that you would like us to clarify further.
> > >
> > > Thank you again for your support and consideration.

---

### Official Review · Reviewer_W5T7 · 2026-02-25

**Soundness:** 3
**Presentation:** 3
**Significance:** 3
**Originality:** 3
**Overall Recommendation:** 4
**Confidence:** 3

**Summary:**

This paper explores a critical yet frequently overlooked challenge faced by embodied agents in closed-loop control: as tasks progress, accumulated context causes the input length for replanning to grow. This growth induces heavy-tailed inference latency, ultimately causing the system to fail in meeting real-time Service-Level Objectives.
To address this, the authors propose BRACE, a controller framework that formulates replanning as a budgeted control problem. They also introduce E-RECAP, a reusable context compression module.
Experiments were conducted across three platforms: Meta Habitat, RoboFactory, and Microsoft AirSim. The results demonstrate that while existing baselines achieve high task success rates, they frequently violate latency constraints (with an SLO violation rate as high as 85.5%). In contrast, combining BRACE with E-RECAP reduces this rate to 4.7% with negligible loss in task success.

**Compliance With Llm Reviewing Policy:**

Affirmed.

**Final Justification:**

The authors’ response has addressed my primary concerns, particularly by adding the missing open-loop baseline and providing additional evidence under more challenging settings. These revisions help clarify the intended contribution and improve the overall completeness of the experimental evaluation.

However, some issues still remain. In particular, the generalization boundary of E-RECAP (e.g., whether retraining or fine-tuning is required under new tasks or significant domain shifts) is not yet fully clarified, and the interpretability of the qualitative visualizations still appears limited.

Therefore, while I appreciate the authors’ efforts and improvements, I am inclined to maintain my original score. I wish the authors the best of luck with their submission.

**Key Questions For Authors:**

See weakness

**Limitations:**

Lack of impact statement

**Strengths And Weaknesses:**

**Strengths**
- The paper acutely points out that current embodied AI research overly focuses on "Success Rate" while neglecting system-level stability . The authors demonstrate that high success rates can mask severe real-time failures (i.e., deadline misses) . This perspective of treating replanning costs and tail latency as first-class objects of study is crucial for transitioning from the laboratory to real-world deployment .
- The paper employs a rigorous experimental design and baseline comparison:
*Cross-domain validation*: Experiments cover three distinct domains: navigation, manipulation, and multi-agent coordination, demonstrating the method's generality .
*Budget-matched baselines*: Instead of simply comparing "compressed vs. uncompressed," the authors established "budget-matched" baselines. This strongly proves that E-RECAP's performance improvement is not merely because "tokens are fewer," but because it preserves more critical information .

**Weaknesses**
- *Insufficient experimental results.* Table 2 in the experimental section only presents the "No BRACE" baseline. How does a baseline perform without any replanning module (i.e., open-loop execution)? Additionally, could the authors provide performance data for BRACE on datasets with lower initial success rates to demonstrate its efficacy in more challenging scenarios?
- *Insufficient discussion on the training cost and generalization of E-RECAP.* What is the specific latency overhead introduced by the E-RECAP module itself (i.e., the lightweight predictor and the pruning operation)? Since E-RECAP requires training a predictor to score tokens, is retraining or fine-tuning this pruning module necessary when deploying to new tasks? Does the pre-trained pruner remain effective under significant domain shifts (e.g., moving from indoor navigation to open-world UAV scenarios)?
- *Lack of interpretability in visualizations.* The visualization experiments presented in the paper lack clear interpretability, making it difficult to extract effective information from the figures.

---

> ### Author Rebuttal · Authors · 2026-03-31
>
> Thank you for the helpful comments, Reviewer **W5T7**.
>
> The revised paper is available at **https://anonymous-2026.github.io/BRACE-ICML/static/paper.pdf**
>
> The anonymous project website is at **https://anonymous-2026.github.io/BRACE-ICML**
>
> The updated experiments, tables, videos, code, supplementary assets, and blue-marked revisions are reflected there.
>
>
> ### 1. Open-loop baseline and harder setting
>
> > Note
> > The concern is that the paper should include a true open-loop baseline and also show whether BRACE still matters in a harder regime.
>
> We agree with both parts of this concern, and both are now addressed directly in the revised main paper. In the Experiments section, we added a compact open-loop and harder-setting evidence snapshot. On the retained Habitat PointNav plus shortest-path-noise slice, a true no-initial-plan or no-replanning baseline is now reported separately from No BRACE. On the retained RoboFactory Pass-Shoe harder-setting, the difference is much sharper: open-loop, frozen plan, and No BRACE all fail badly, while BRACE plus E-RECAP retains both recovery and deadline control.
>
> | Domain | Method | Task metric | Lat P95 | SLO viol. |
> |---|---|---|---:|---:|
> | Habitat | No-initial-plan | 0.519 SPL | -- | N/A |
> | Habitat | No BRACE | 0.515 SPL | 5492 ms | 93.4% |
> | Habitat | **BRACE + E-RECAP** | 0.519 SPL | **2486 ms** | **0.0%** |
> | RoboFactory | Open-loop | 0.0% | -- | -- |
> | RoboFactory | Frozen plan | 0.0% | 72.8 ms | 0.0% |
> | RoboFactory | No BRACE | 0.0% | 312.7 ms | 27.6% |
> | RoboFactory | **BRACE + E-RECAP** | **80.0%** | **247.2 ms** | **4.6%** |
>
> This snapshot is now in the main paper so that the reader can see, without going to the appendix first, that the missing baseline has been added and that the harder-setting evidence is not merely a supplementary side note.
>
> ### 2. Overhead and transfer boundary of E-RECAP
>
> > Note
> > The concern is that the paper should state the cost of E-RECAP more directly and avoid overstating transfer beyond the current evidence.
>
> We agree that the earlier paper did not surface E-RECAP-side overhead clearly enough. The revised appendix now separates this explicitly with a Habitat phase-overhead table and a direct replanning-effect table. This makes the controller cost, pruning-side cost, and schedulability difference visible without forcing the reader to infer them only from final task success. At the same time, we keep the transfer claim scoped appropriately: the revised paper includes broader backbone, method, and manipulation-platform evidence, but it does not claim a theorem-level guarantee of zero-shot transfer under arbitrary domain shift.
>
> The revised paper therefore says both what the current evidence supports and what it does not yet support. This is helpful for the rebuttal because it answers the reviewer directly without overstating the scope of the new experiments.
>
> The phase-overhead table can be distilled into the numeric summary below.
>
> | Variant | Prune mean | Planner mean | End-to-end P95 | Tok. after |
> |---|---:|---:|---:|---:|
> | No BRACE | 0.76 | 3804.74 | 5492.08 | 256.85 |
> | Pruning only | 37.21 | 2414.07 | 2488.24 | 20.21 |
> | BRACE + E-RECAP | 35.63 | 2414.05 | 2486.31 | 20.21 |
> | Recency truncation B20 | 0.69 | 2287.56 | 2289.53 | 20.00 |
>
> ### 3. Visualization interpretability
>
> > Note
> > The concern is that the earlier qualitative figures were difficult to interpret and did not clearly support the quantitative claims.
>
> We agree that the earlier qualitative figures were not self-explanatory enough. We therefore changed the qualitative assets inside the paper itself rather than only pointing to the website. In the revised main paper, the AirSim qualitative figure is replaced by a retained storyboard-style comparison, and the main text also adds a RoboSuite retained failure case beyond the original RoboFactory anchor. In the appendix, we removed the weaker Habitat supplementary qualitative comparison, retained TakePhoto as a supplementary success storyboard, added a compact Two-Arm Handover storyboard, and added a LIBERO success-versus-baseline pair.
>
> This change is paired with the revised simulation coverage table in the appendix, which now makes the overall platform, task, and scene coverage explicit rather than leaving the broader evaluation footprint implicit.
>
> ### 4. Impact statement
>
> > Note
> > The suggestion is to add an impact statement to the paper.
>
> We accepted this reminder. The revised paper now includes the impact discussion.

---

> > ### Author Rebuttal · Reviewer_W5T7 · 2026-04-02
> >
> > Thanks for the response. I do not have further concerns.

---

> > > ### Author Response · Authors · 2026-04-03
> > >
> > > Dear Reviewer W5T7,
> > >
> > > Thank you again for your thoughtful feedback and for confirming that your concerns have been addressed. We are glad that our rebuttal helped clarify the key points.
> > >
> > > If there are still any aspects that you feel could benefit from further clarification, please feel free to let us know — we would be happy to elaborate. Otherwise, we wanted to ask whether you might be open to providing a more positive final evaluation based on your current assessment of the work.
> > >
> > > Thank you for your time and support.

---

### Official Review · Reviewer_awgN · 2026-02-28

**Soundness:** 3
**Presentation:** 3
**Significance:** 3
**Originality:** 3
**Overall Recommendation:** 4
**Confidence:** 3

**Summary:**

The paper develops a framework named BRACE that decides whether replanning is required based on current resources. In particular, it is a budgeted replanning framework that treats each planner invocation in embodied agents as a controlled, metered systems primitive with explicit token and latency budgets, rather than an unbounded LLM call. At each trigger, BRACE decides whether to replan, selects a replanning mode, and allocates a per-call token budget and service-level latency objective, while incorporating stabilization policies and phase-level audit logging to make tail latency and SLO violations measurable and comparable. As a composable efficiency module, this work introduces E-RECAP, which predicts token importance from intermediate transformer states and prunes long replanning contexts across layers while preserving critical head and tail tokens. Across navigation, manipulation, and multi-agent benchmarks, the system reduces replanning tokens and tail latency, lowers SLO violation rates, while maintaining task success rate.

**Compliance With Llm Reviewing Policy:**

Affirmed.

**Final Justification:**

The rebuttal has addressed my concerns and I will remain my recommendation toward acceptance.

**Key Questions For Authors:**

1. While the empirical results are convincing, the framework appears primarily motivated by systems intuition. Do you have any theoretical analysis or formal guarantees regarding stability, bounded SLO violations, or optimal budget allocation under BRACE? For example, can the controller be analyzed from a control-theoretic perspective, or are there conditions under which SLO violation rates can be bounded theoretically?

2. BRACE may block replanning through cooldown/commit windows unless failure-aware overrides are triggered. In cases where replanning is truly required but the controller suppresses it, how do you ensure that performance or safety is not degraded? Is there empirical evidence quantifying cases where replanning was skipped but would have been beneficial?

3. In multi-agent settings, does BRACE assume a centralized controller and shared replanning context buffer? If so, how would the framework extend to decentralized agents with independent planning loops and communication delays? Are per-agent budgets or distributed SLOs possible within your design?

**Limitations:**

While the paper includes a systems-oriented discussion of limitations (e.g., overhead trade-offs, pruning effects, stability concerns), it does not address potential negative societal impacts. Given that the work targets embodied agents operating in real or semi-real environments (navigation, manipulation, multi-agent traffic), a brief discussion would strengthen the paper.

Some discussions would be useful:

- Show a case study where skipping or delaying replanning and potential negative impact.

- Discuss potential deployment in surveillance, military robotics, or traffic control systems, where improvements in replanning efficiency could enable large-scale automation with societal consequences.

**Strengths And Weaknesses:**

### Strengths

- The paper addresses a realistic systems bottleneck in embodied LLM-based agents, replanning latency under context growth, which has clear practical implications for real-time deployment.

- The proposed framework, efficiency module, accounting protocol, and experimental setup are presented in an organized and easy-to-follow manner.

- The evaluation spans three embodied platforms (navigation, manipulation, multi-agent traffic) and includes tail latency, SLO violation rates, and ablations, providing convincing empirical support beyond simple success metrics.

- The paper contributes a measurement perspective: reporting replanning cost at the call level, including tail percentiles and SLO violation rates with phase-level accounting. This provides an alternative evaluation standard, in addition to average success/latency, using stability-aware metrics.

### Weaknesses

1. The framework is primarily justified through systems intuition and empirical evidence, but there is no formal analysis of stability, optimality, or probabilistic guarantees on SLO violation rates. The controller mechanisms (cooldown, commit windows, failure overrides) are heuristic, and the paper does not provide control-theoretic, queueing-theoretic, or complexity-based analysis explaining when or why the closed-loop system is provably stable.

2. *Correct me if I'm wrong on this point.* BRACE may block replanning due to cooldown or commit constraints. In situations where replanning is urgently required (e.g., safety-critical events or rapid environmental changes), suppressing replanning could degrade performance or safety. Similarly, if a replanned policy is suboptimal but the commit window prevents immediate correction, the system may persist with a poor plan.

3. The framework introduces multiple additional hyperparameters (e.g., cooldown threshold, commit window length, token budget, SLO target, pruning keep ratios, failure override rules). The effectiveness and stability of BRACE likely depend on careful tuning of these parameters, yet the paper does not provide systematic sensitivity analysis or guidance for adapting them to new environments. This may limit robustness and ease of deployment in diverse real-world settings.

---

> ### Author Rebuttal · Authors · 2026-03-31
>
> Thank you for the helpful comments, Reviewer **awgN**.
>
> The revised paper is available at **https://anonymous-2026.github.io/BRACE-ICML/static/paper.pdf**
>
> The anonymous project website is at **https://anonymous-2026.github.io/BRACE-ICML**
>
> The updated experiments, tables, videos, code, supplementary assets, and blue-marked revisions are reflected there.
>
> ### 1. Formal scope and guarantees
>
> > Note
> > The concern is that the original paper did not make the formal scope and guarantee level sufficiently explicit.
>
> We agree that the original submission did not state its formal boundary clearly enough. In the revised paper, right after the BRACE gate equation in the Method section, we now state a deterministic anti-churn property: without failure-aware overrides, consecutive replanning calls must be separated by at least delta controller steps, and accepted plan replacements must survive at least omega steps before another replacement is admitted. We then restate this in the appendix Controller sweep section and add a short proof sketch. We present this as controller-side structure and auditability, not as a full theorem for the entire embodied closed loop.
>
> This change now appears in two paper locations: the Method section states the property next to the gate equation, and the appendix Controller sweep section gives the proof sketch and links it to the controller-sweep evidence. This makes the formal scope explicit without changing the core claim of the paper.
>
> ### 2. Whether useful replans are suppressed
>
> > Note
> > The concern is that cooldown or commit may block replanning in cases where replanning was actually beneficial.
>
> We agree that this should be answered empirically rather than only by intuition. The revised appendix now includes a suppressed-trigger outcome audit that classifies withheld triggers as recovered, harmless, or harmful within a short horizon. The result is that most suppressed triggers reduce churn without causing harm, and harmful cases are rare rather than dominant.
>
> | Gate reason | Recovered | Harmless | Harmful |
> |---|---:|---:|---:|
> | commit window | 13/13 | 0/13 | 0/13 |
> | cooldown | 23/34 | 9/34 | 2/34 |
>
> This table is now reported in the appendix, and the revised AirSim discussion points to it directly.
>
> This is useful for the rebuttal because it replaces an intuition-only argument with a measurable audit. Instead of claiming in words that the gate is usually safe, we now show what happens after withheld triggers and make clear that harmful suppressions are rare rather than typical.
>
> ### 3. Parameters and deployment guidance
>
> > Note
> > The concern is that the controller may depend on many tuned hyperparameters without enough deployment-facing guidance.
>
> We agree that the earlier version did not summarize the parameter roles clearly enough. The revised paper now ties this discussion to the per-domain setup table and the controller sweep. We explicitly separate controller knobs, efficiency knobs, and deployment targets, and we clarify that the present multi-agent experiments use centralized or shared accounting. Per-agent budgets and distributed SLOs are now described as a direct extension path rather than an already validated claim.
>
> For clarity, the revised paper now distributes this answer across the setup table, the controller sweep, and the revised discussion text, so the reader can see both the role of each parameter and the current deployment boundary.
>
> The revised paper can also be distilled into the numeric controller-sweep snapshot below, taken directly from the appendix sweep on the proxy environment.
>
> | Variant | delta | omega | w | Success | Calls/ep | Deadlocks/ep |
> |---|---:|---:|---:|---:|---:|---:|
> | No BRACE + E-RECAP | n/a | n/a | n/a | 5.0 | 167.18 | 157.53 |
> | BRACE + E-RECAP | 0 | 0 | 1 | 98.3 | 14.37 | 0.00 |
> | BRACE + E-RECAP | 0 | 2 | 1 | 75.0 | 24.40 | 14.78 |
> | BRACE + E-RECAP | 3 | 2 | 1 | 78.3 | 21.78 | 12.12 |
>
> ### 4. Impact statement
>
> > Note
> > The suggestion is to add a brief discussion of broader impact and deployment risk.
>
> We accepted this point. The revised paper now includes the impact discussion.

---

> > ### Author Rebuttal · Reviewer_awgN · 2026-04-01
> >
> > Thanks for the response. I do not have further concerns.

---

> > > ### Author Response · Authors · 2026-04-03
> > >
> > > Dear Reviewer awgN,
> > >
> > > Thank you for your constructive feedback and for indicating that your concerns have been addressed. We sincerely appreciate your updated assessment and the time you devoted to carefully reviewing our responses.
> > >
> > > If there is anything that still requires further clarification, we would be happy to provide additional details. Given your positive updated view, may we kindly ask whether you would consider giving a more positive final evaluation to the paper?
> > >
> > > Thank you again for your consideration.

---

### Decision · Program_Chairs · 2026-04-30

**Decision:**

Accept (regular)

**Comment:**

This paper addresses an important and under-studied problem in deploying LLM-based decision makers for embodied agents, specifically in devoting resources to replanning.

The reviewers are in agreement that this is an interesting and important problem, and generally are accepting of the proposed solution.
The main weakness is that in its present form, the approach is heuristic-driven and missing a clear formalism. Nevertheless, it appears to be a reasonable initial contribution, and should enable future work in its path.
There are questions about details on implementation, design tradeoffs, and hand-engineering, that should be addressed in the revision.

While the reviewers did not raise this point, I see clear and strong connections between this work and the long-established research space on metareasoning for planning and resource-bounded rationality, which have been investigated for decades. The authors should review the previous work in this field to better inform their work, and to better position their research and draw from previous knowledge. An excellent such resource is the paper on "Principles of Metareasoning", Stuart Russell and Eric Wefald: https://doi.org/10.1016/0004-3702(91)90015-C
Another relevant paper is "Metareasoning for Planning Under Uncertainty" by Christopher H. Lin, Andrey Kolobov, Ece Kamar, Eric Horvitz: https://www.ijcai.org/Proceedings/15/Papers/229.pdf